# Roads Detection and Parametrization in Integrated BIM-GIS Using LiDAR

**Luigi Barazzetti \*** , **Mattia Previtali and Marco Scaioni**

Department of Architecture, Built environment and Construction engineering (ABC), Politecnico di Milano, Via Ponzio 31, 20133 Milan, Italy; mattia.previtali@polimi.it (M.P.); marco.scaioni@polimi.it (M.S.)
**\*** Correspondence: luigi.barazzetti@polimi.it; Tel.: +39-02-2399-8779

**Abstract:** Building Information Modeling (BIM) has a crucial role in smart road applications, not only limited to the design and construction stages, but also to traffic monitoring, autonomous vehicle navigation, road condition assessment, and real-time data delivery to drivers, among others. Point clouds collected through LiDAR are a powerful solution to capture as-built conditions, notwithstanding the lack of commercial tools able to automatically reconstruct road geometry in a BIM environment. This paper illustrates a two-step procedure in which roads are automatically detected and classified, providing GIS layers with basic road geometry that are turned into parametric BIM objects. The proposed system is an integrated BIM-GIS with a structure based on multiple proposals, in which a single project file can handle different versions of the model using a variable level of detail. The model is also refined by adding parametric elements for buildings and vegetation. Input data for the integrated BIM-GIS can also be existing cartographic layers or outputs generated with algorithms able to handle LiDAR data. This makes the generation of the BIM-GIS more flexible and not limited to the use of specific algorithms for point cloud processing.

**Keywords:** building information modelling; GIS; infrastructure; LiDAR; point cloud; smart roads

## 1. Introduction

Smart roads can be defined as roads coupled with digital information able to provide capabilities for advanced applications such as traffic monitoring, real-time data delivery to drivers, improved safety conditions, or support to the development of connected and autonomous vehicles (CAVs), among others [1–6]. The need for advanced models to support the integration of multi-source digital data makes Building Information Modeling (BIM) a fundamental tool for different specialists in different fields [7]. Smart road projects can also be related to BIM, but they also require multiple data sources coming from different domains, among which Geographic Information (GI) and digital surveying technologies. For instance, photogrammetry and laser scanning techniques (from satellite, airplanes, drones, or ground-based platforms) are the primary source of metric information for projects in wide areas. Existing or newly generated Geographic Information System (GIS) layers provide metric data with associated geospatial information, revealing previous or current conditions.

In recent years, BIM has proved to be a valid tool to support projects in the architecture and construction industry. Most BIM software packages available on the commercial market are mainly used for buildings and their components. The typical procedures for the generation of as-designed BIM require the assembly of predefined components already subdivided into families, such as walls, columns, floors, roofs, windows, doors, ceilings, and ramps. Different object libraries are available on the web. Alternatively, the user can generate tailored parametric objects for a specific application. Nowadays, the creation of a BIM can be carried out with several software packages (e.g., Autodesk

Revit®, ArchiCAD®, AECOsim Building Designer®, Tekla BIMsight®, etc.), which allow users to virtually assemble the building [8].

BIM projects are not limited to new constructions. Interventions on existing buildings (such as reuse, rehabilitation, or restoration) require the generation of a metrically accurate as-built BIM [9]. A detailed survey with modern digital sensors for metric documentation is one of the main inputs. The need for accurate geometric information to support as-built BIM projects is also confirmed by the growing interest in image- or point-cloud-based processing techniques in BIM software. For instance, Autodesk® has integrated into several software packages (e.g., AutoCAD®, Revit®, Civil 3D®) the opportunity to handle point clouds through the Recap 360® module, which can process laser scanning data and generate photogrammetric point clouds from images. Recap® can also register laser scans via "cloud-to-cloud" algorithms (iterative closest point, ICP [10]) or generate point clouds from a set of images using photogrammetric/structure-from-motion algorithms [11]. Bentley ContextCapture® offers automatic procedures for the creation of dense point clouds from huge blocks of images, such as the ones collected during airborne photogrammetry and drone missions. ContextCapture® can also import and integrate laser scanning point clouds in the automatic processing workflow.

Although BIM was mainly developed for buildings, concepts can be extended to infrastructures, starting from the design to the construction and serviceability phases. According to [12], applications at the scale of infrastructure includes five main categories, as shown in Table 1. An extensive review of BIM for infrastructures is also reported in [13–15].

**Table 1.** Main application fields of Building Information Modeling (BIM) concept at infrastructure scale.

| Main Field | Application Examples |
|---|---|
| Transportation infrastructure | - Roads<br>- Railways<br>- Bridges<br>- Tunnels<br>- Mass transit hubs (such as airports, ports & harbors) |
| Energy infrastructure | - Power generation plants (nuclear, wind, tidal, etc.)<br>- Oil and gas (storage/distribution terminals, refineries, wells)<br>- Mining |
| Utility infrastructure | - Networks/pipelines for the delivery and removal of electricity, gas, water and sewage |
| Recreational facilities infrastructure | - Parks<br>- Stadiums |
| Environmental infrastructures | - Dams<br>- Levees<br>- Weirs or embankments |

This paper describes BIM extended to road projects. The main idea is the development of a procedure able to provide an advanced three-dimensional (3D) model of road networks coupled with existing or new geospatial information. We also introduce a method for road detection and extraction from Light Detection And Ranging (LiDAR) point clouds, which is the input in the workflow proposed in the paper. On the other hand, the proposed procedure is not limited to data generated from LiDAR using the implemented algorithm. Other methods available in the technical literature (or in commercial software) can be used to generate a vector-based representation of the road profile. Moreover, existing cartographic datasets can be used as well. The generation of the BIM-GIS is, therefore, flexible. It just relies on the availability of such data, and it is independent of the procedure used for their generation.

An advanced 3D BIM of a road network contains information about the geometry of the road and its components, such as the longitudinal profile, transversal sections, gutters, drainage systems, railings, crash cushions, signals, etc. The model of the infrastructure is registered in a mapping reference system through georeferenced geospatial data such as digital terrain/surface models (DTM/DSM), orthophotos,

vector-based geospatial databases, GNSS (Global Navigation Satellite System) measurements, and other georeferenced 3D models.

Modeling must be carried out using a parametric approach, which differs from traditional "pure" modeling methods that aim at reconstructing only the geometry using static surfaces or solids. Eastman [16] defined the following requirement of parametric objects for buildings, which can be extended to road components:

- To contain geometric information and associated data and rules;
- To have non-redundant geometry, which allows for no inconsistencies;
- To have parametric rules that automatically modify associated geometries when inserted into a building model or when changes are made to associated objects;
- Can be defined at different levels of aggregation;
- Can link or receive, broadcast, or export sets of attributes such as structural materials, acoustic data, energy data, and cost, to other applications and models.

The road can be intended as an assembly of "intelligent objects" that interact, whose dimensions can be modified in a parametric way so that the road database becomes a dynamic system. The use of BIM for road projects must include both newly designed roads and existing ones, which must be modeled beforehand in the digital environment. An example of typical roads available in commercial libraries is shown in Figure 1, in which the software used is Autodesk InfraWorks® (AIF).

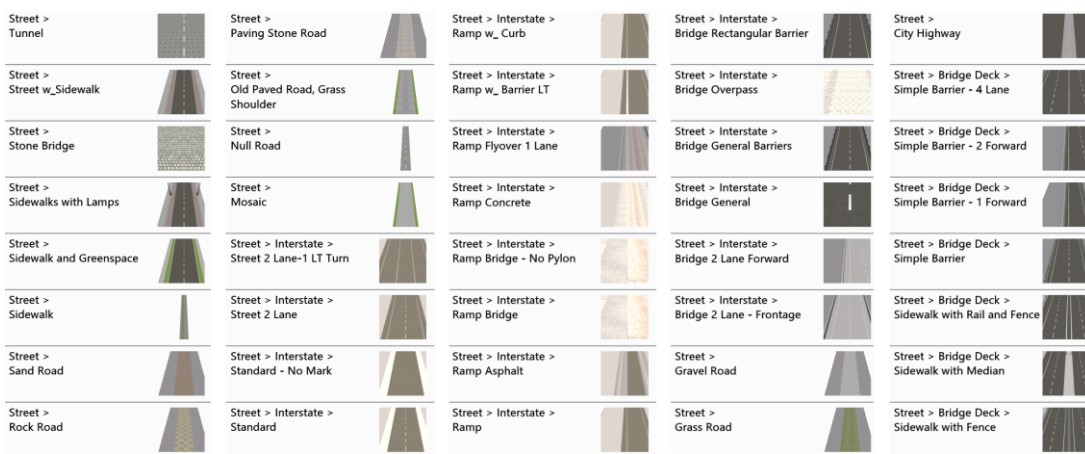

**Figure 1.** Typical road objects available as parametric elements for Building Information Modeling (BIM) in Autodesk InfraWorks® (AIF).

The importance of BIM for infrastructures is also confirmed by the improvements introduced in the Industry Foundation Class (IFC). The current release (IFC4 Add2) has novel tools to support infrastructure projects such as the use of georeferenced coordinate systems, the separation between site and terrain, or the introduction of novel geographic features. The upcoming release IFC5 is expected to include full support for various infrastructure domains and more parametric capabilities. The IFC Alignment Project aims at providing the first IFC format suitable for infrastructure projects, introducing an alignment standardization workflow focused on roads, railways, bridges, and tunnels. Some advanced tools are already available in IFC 4.1 to design linear features using a parametric representation based on curves.

The analysis of roads as BIM objects cannot be limited to the infrastructure scale. Small components are an essential part of roads and should be added in projects requiring a very high level of detail. Some libraries are already available, such as BIMcomponents.com, developed by GRAPHISOFT SE®, with more than 230 road objects (vehicles, traffic signs, barriers . . . ). Highways England® has launched a Digital Component Library (DCL) for highways projects, which contains more than 80 road objects. Figure 2 shows some components in Autodesk InfraWorks®.

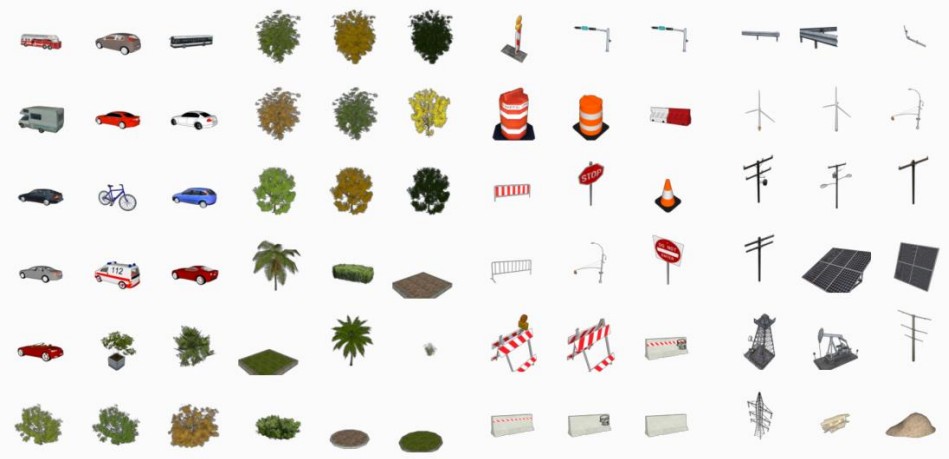

**Figure 2.** Some object components available in AIF.

As mentioned, BIM for road projects can be based on existing information (such as cartographic layers or georeferenced CAD drawings) or integrated by newly acquired metric information. Laser scanning and photogrammetric point clouds are an essential source of metric information to capture as-built conditions. The scan-to-BIM process refers to procedures able to convert point clouds into parametric objects using manual, semi-automatic, or automatic approaches. Nowadays, some BIM software packages can directly import point clouds to facilitate modeling operations. In the case of infrastructures, mobile sensors installed on airborne or terrestrial vehicles [17–21] are one of the most common solutions to capture accurate and dense point clouds.

Figure 3 shows an example of LiDAR data imported in a BIM-GIS environment (AIF). Data can be inspected and modified using the point cloud, without requiring extra work in third party software. The approach proposed in this paper is different. The point cloud is imported in parametric software to refine the results provided by an automatic road extraction algorithm, which gives a reconstruction based on parametric objects that can be modified without remodeling. The point cloud is, therefore, exploited at the beginning of the process to obtain parametric objects. Then, it is used for a manual inspection of the results obtained with the automatic procedure. Although the automatic extraction of the road is carried out with an automatic detection procedure (described in Section 4), BIM-GIS generation requires only a set of shapefiles as input. Thus, existing cartographic information can be directly used (if available). Alternatively, other algorithms for road extraction using LiDAR can be used to generate the input for the integrated BIM-GIS environment.

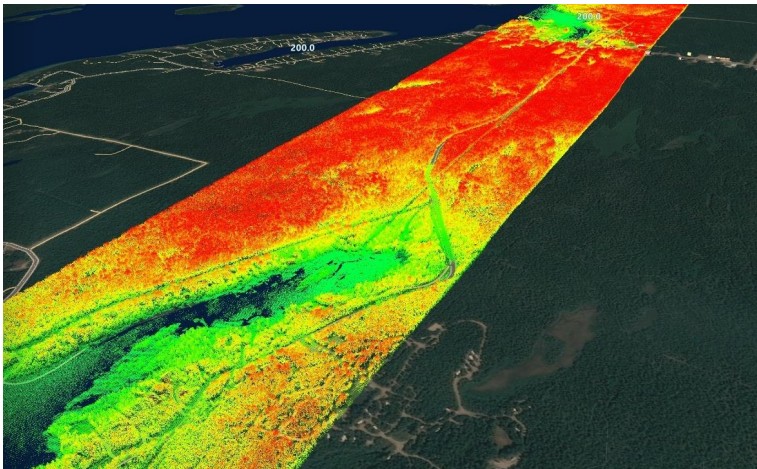

**Figure 3.** The point cloud imported in parametric modeling software based on an integrated BIM/GIS environment (AIF).

## 2. Overview of Automatic Road Extraction Methods from LiDAR Data

Automated detection and extraction of roads [22] have been an active research field in different scientific domains with different approaches. Nowadays, remote-sensing techniques provide different input data such as multispectral, hyperspectral, and panchromatic images, radar imagery, and LiDAR [23,24]. Road extraction using optical methods is generally inefficient in highly vegetated areas since trees may compromise the visibility of the road surface using passive sensors [25]. Active sensors, such as LiDAR, exhibit a higher potential than passive remote sensing [26] since beams can penetrate canopies. The availability of range data allows one to distinguish between roads and features presenting similar radiometric/spectral signature (e.g., different types of roofs). On the other hand, bare earth may exhibit range and intensity values similar to roads [27], resulting in a difficult separation of these features. Consequently, the detection of road borders can be less accurate than in the case of optical data. LiDAR data present a higher potential due to the lack of sensitivity to shadows and the reduced influence caused by occlusions.

The following overview of road extraction will mainly focus on papers presenting aerial-based LiDAR systems since the paper deals with this typology of data. However, a brief paragraph is also devoted to road extraction starting from mobile LiDAR on terrestrial mapping vehicles.

One of the first approaches developed for automatic road extraction from LiDAR data [28] started from a Digital Surface Model (DSM) directly derived from LiDAR. Road candidate points were obtained by filtering using a threshold distance from the Digital Terrain Model (DTM) and intensity value. Road patches were later connected into a road network. Zhao and You [29] developed a method based on template matching. In the first stage, LiDAR points are classified as ground and off-ground elements. In a second step, a template schema is used to search for roads on ground intensity images. Road widths and orientations are determined by a subsequent voting scheme. Hu et al. [30] presented a combination of three algorithms. First, Mean Shift Clustering (MSC) is used to detect road center points. As a second step, the Tensor Voting method is applied to highlight the main linear features, and then a Hough Transform allows us to detect road centerlines. A semi-automatic approach is presented in [31]. First, a set of road seeds is manually selected. Then, starting from those seeds, a region-growing algorithm is used to define road areas. In a third stage, fast parallel thinning is used to extract road centerlines. The manual modification of the seed can remove incorrectly extracted roads. Hui et al. [32] developed a method composed of three key algorithms: Skewness balancing, Rotating neighborhood, and Hierarchical fusion and optimization (SRH). Skewness balancing is used to define an optimal intensity threshold for road points. Narrow streets are then removed with a rotating neighborhood algorithm while road centerlines are derived by using hierarchical fusion and optimization techniques.

Another combination of algorithms for road centerline extraction is presented in [33]. Here, the three main steps are: (i) the detection of the road center point using adaptive MSC, (ii) local principal component analysis for extracting linear distributed points, and (iii) hierarchical grouping for connecting primitives into complete roads network. Tejenaki et al. [34] found road centerlines by combining intensity data and normalized DSM. Mean Shift is used to filter intensity data, which are combined with different normalized DSM products to minimize the effects due to large parking lots and similar areas. In the final stage, road centerlines are extracted by using a Voronoi-diagram-based approach and then by removing dangle lines. Liu et al. [35] specifically focused on mapping roads that are used for timber transport. In this work, a Dense Dilated Convolutions Merging Network (DDCM-Net) is used to detect roads in LiDAR data.

In recent years, road extraction based on car-based Mobile Laser Scanning (MLS) has become more popular for road management and inspection, pavement condition control, and the generation of road inventories [36]. A mobile laser scanner is mounted on a vehicle moving along the road. The acquisition system is generally coupled with GNSS (Global Navigation Satellite Systems) and INS (Inertial Navigation Systems) to calculate position and attitude. The high accuracy and density of the point cloud can be used for the extraction of road information. More specifically, the road surface is

extracted with best-fit plane detection algorithms based on robust methods (e.g., RANSAC) or using Principal Component Analysis (PCA) [37]. Road shoulders [38], sidewalks, or road alignment [39] can be extracted using decision trees [40] or fusion of feature-based and deep learning approaches [41]. Another important topic is road marking extraction. Detection of pavement markings is generally carried out by exploiting LiDAR intensity information [42,43]. In Crosilla et al. [44], LiDAR data are classified based on both elevation and intensity information by exploiting skewness and kurtosis analysis.

Some general aspects may be highlighted from this short overview of the existing literature. First, although LiDAR intensity is a fundamental aspect for identifying roads (see also [44]), the separation between road points and bare soil points is still problematic. Moreover, different roads may present multiple intensity values connected to the different construction materials (e.g., asphalt, concrete, etc.). A second problematic aspect of the automated method is connected to the influence of the attached areas (e.g., parking lots). Irregular point distribution and variations of road pattern/width may also affect the efficiency of road detection.

## 3. The Workflow of the Proposed Method

The proposed workflow is a multi-step procedure that allows one to classify airborne LiDAR data, generate a vector-based representation for linear or polygonal features, add more geospatial (GIS) data from online repositories, and then integrate all the information in the AIF BIM-GIS environment. The entire workflow (Figure 4) starts from multiple input data with different intermediate processing levels. For instance, the LiDAR point cloud is processed by an automated classification and filtering method (Section 4), obtaining a set of georeferenced vector files. The use of the proposed algorithm for road extraction is not mandatory. Existing cartographic information or other methods for road detection from LiDAR can be used, offering the opportunity to directly focus on the generation of the integrated BIM-GIS, without the need of specific algorithms for the creation of the input. Moreover, other data, such as DTM and orthophotos of the considered area, can be retrieved from repositories. The idea of the proposed method is to reuse as much as possible existing information, whose quality can be tested and refined with LiDAR data.

OpenStreetMap (OSM, www.openstreetmap.org) was also used to collect additional geospatial data for the considered area. Section 5 describes how some BIM/GIS software packages can have direct access to OSM layers, which can be imported and turned into parametric BIM objects. On the other hand, the accuracy and completeness of these models could be insufficient for practical applications. In the proposed workflow, OSM data are used as visual support to refine the results of the LiDAR processing procedure.

The Multispectral LiDAR Data Set 2 "Tobermory" provided by ISPRS Working Group III/5 [27] was used in this research. This data set covers a natural coastal area located in Tobermory (Ontario, Canada). The Optech Titan multispectral Airborne Laser Scanning (ALS) sensor was used to acquire data in April 2015, at a flying height of approximately 460 m a.s.l. and a speed of 140 knots. The Optech Titan is equipped with three active beams that acquire independent data at three wavelengths: 532 nm, 1064 nm, and 1550 nm. Multispectral LiDAR data increase the possibility of feature classification since more spectral properties can be evaluated if compared to single monochromatic ALS systems. The data set consists of 11 strips over an area of approximately 10 km × 2 km and a density of approximately 12 points/m$^2$. Only seven strips were processed.

LiDAR data are segmented into three distinct categories: ground, trees, and buildings. This classification is carried out using the Random Forest classifier. At this stage, "ground" points mainly contain roads, parking lots, bare ground, low-land grass, etc. The next step is the identification of candidate "road" points. This task is accomplished within a four-step approach: (i) linearity is computed at different scales per each "ground" point; (ii) Mean Shift is used to cluster points and classify them as belonging to a road; (iii) detected points are used to robustly fit piecewise linear features; and finally (iv) road segments are grouped and merged to create the road network. Points classified as "tree" and "building" can be used to give context to the data. For example, "tree" points

can be used to create a vegetated area in a parametric way, while "building" points can be used to generate parametric buildings. An example of data contextualization is provided in Section 5.

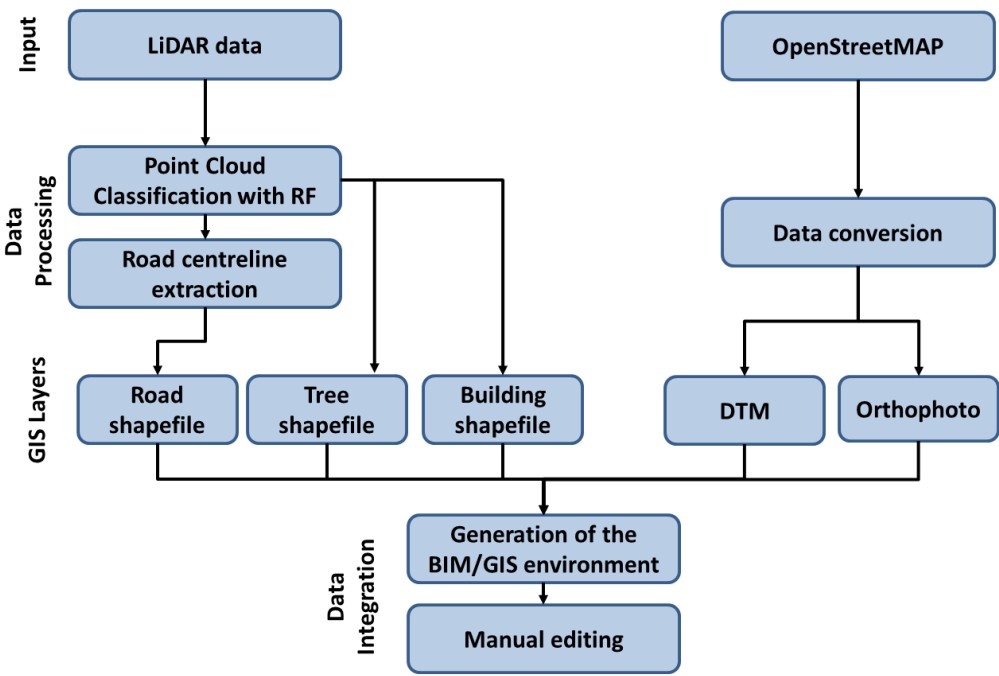

**Figure 4.** Schematic workflow of the proposed procedure: from input data to the production of an integrated BIM/GIS. RF: Random Forest; DTM: Digital Terrain Model.

The last part of the workflow is related to the BIM parameterization of GIS layers, in which roads, buildings, and vegetation are turned into BIM objects. The level of parameterization depends on the characteristics of each element. Particular attention is paid to roads, which can be detected, classified, and modeled using predefined BIM typologies. Different families of roads for the considered area can be generated using BIM. A distinction based on primary and country roads is performed to take into consideration the actual type of the road and different levels of detail for both geometry and semantic information.

First, a road can be a "standard" road, a bridge, or a tunnel. Bridges are considered as other elements with different parameterization levels (e.g., height, number of pillars, etc.), notwithstanding that bridges can be assimilated to roads during several analyses (e.g., traffic flow). Then, every road typology has predefined materials associated with its different parts as well as specific components (e.g., traffic lights, signals, etc.). Geometric parametrization is carried out on those parameters that can be parametrically modified so that a modification of the geometric model is reflected in the database, and vice versa. Some geometric parameters that can be modified, among others, are track width, track inner offset height, track outer offset height, transition zone size, etc. The road will be therefore modeled using its profile, which is turned into a parametric BIM road element after assigning the specific road typology.

It is also possible to create asymmetric roads. Usually, one side of the road mirrors the other, but manual editing allows one to handle special situations. The number of forwarding driving lanes can be different from the number of backward driving lanes. After setting the number of lanes, the overall geometry of the road is updated in the model.

Finally, the BIM/GIS model can be integrated with vegetation and buildings. In this research, an approximate model is used for both elements, whereas roads have a superior level of detail. Although the aim is the generation of an accurate and complete model revealing as-built roads, advanced BIM operations may require other elements to obtain a more realistic result. This is the case

for numerical traffic simulation, where buildings and vegetation provide additional occlusions and have a significant impact during the analysis. As the procedure of LiDAR classification can detect vegetation and buildings, both elements were integrated into the proposed workflow to obtain a simplified model. More research work will be necessary to refine these elements, but this is out of the scope of this paper. The overall idea is a flexible solution in which the generation of the BIM-GIS can be carried out using multiple input datasets with variable levels of details, resulting in multiple project proposals without requiring the use of specific algorithms to generate the input for the BIM-GIS model.

## 4. Generation of Road GIS Layers from LiDAR Data

Although AIF provides instruments for downloading OpenStreetMap (OSM) data, these can be outdated, incomplete, or not accurate enough. In the case of the area of Tobermory, only the main roads are available, whereas secondary roads are completely missing. Moreover, some road intersections are inaccurate or wrong. Inaccuracies in the road data network are not acceptable for several types of analyses, e.g., visibility analysis between vehicles at road intersections or investigation of local traffic conditions. LiDAR data can be used to integrate missing information. This section describes the developed methodology for road network identification using LiDAR. As mentioned in the previous sections, the use of the implemented algorithms is not strictly necessary for the generation of a BIM-GIS. The user can use commercial software to implement an ad-hoc algorithm to cope with specific conditions. The authors' idea is to present the used implementation concentrating on both the algorithmic aspects and the output data formats, which are the input for the creation of the integrated BIM-GIS. Thus, users interested in a BIM-GIS in AIF can readapt their algorithms using the output formats here presented.

Figure 5 illustrates the general workflow of the proposed road detection and extraction method from LiDAR data. The proposed method is a multi-level procedure, which is exploited using a bottom-up approach. A difference between the proposed approach and previous research on road centerline extraction can be summarized as follows:

- The multi-scale approach takes advantage of point cloud information at local and global scales, allowing the identification of both main and dirt roads characterized by different size;
- A voting scheme is used to identify and discriminate between roads and other surfaces (parking lots, squares, etc.) characterized by similar local features (local planarity, LiDAR intensity, etc.);
- A regularization procedure generates a road network presenting consistent topological relationships, extending the centerline extraction procedure based on cell complex presented in [45].

Moreover, the paper illustrates a novel procedure for building footprint regularization. The next subsections will thoroughly illustrate the different steps of the proposed approach.

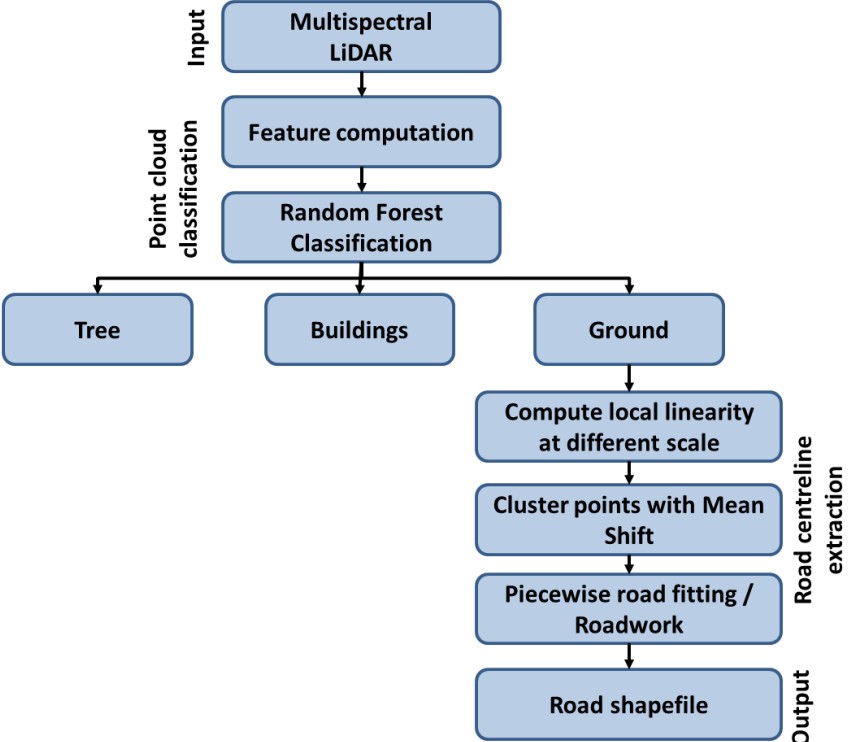

**Figure 5.** Schematic workflow of the proposed method for automatic extraction of road centerline form LiDAR data.

## 4.1. Point Cloud Classification

The first step of the presented workflow is the classification of the point cloud into a set of classes: "ground," "building," and "tree." Nowadays, several workflows exist for point cloud classification based on decision trees, random forest, random fields, and deep learning. The development of a new classification approach is out of the scope of this paper. Here, we used a common classification method and we built on the top of it a procedure for road centerline extraction. The presented classification was carried out using the Random Forest (RF) classifier [46], which yields multiple decision trees using a randomly selected subset of training samples and variables. In other words, RF randomly and iteratively samples the selected variables to generate a large set (a "forest") that represents the statistical behavior of numerous decision trees. A voting strategy was adopted to combine the votes over the constructed trees, while a bagging strategy was used to create training sets from the original data. Bagging randomly selects about two-thirds of the samples from training data to train the trees. Then, the remaining samples, generally called "out-of-bag" (OOB), were used for cross-validation and calculation of the classification error.

The number of decision trees (in RF) was set to 500. Features computed per each LiDAR point can be classified as (i) height-based, (ii) eigenvalue-based, (iii) local plane-based [38], and were intensity-based at 532 nm, 1064 nm, and 1550 nm, respectively. Features were selected according to the most used in point cloud classification, as proposed in [47–49].

The "height-based" (i) features used for classification are:

- $\Delta z$: height difference between the point and the lowest point found in a cylindrical volume whose radius has been empirically set to 7 m (discrimination between "ground" and "off-ground" points);
- $\sigma_z^2$: the height variance computed for *k*-nearest neighboring points, where *k* has been empirically set to 50.

The "eigenvalues-based" (ii) features come from the variance-covariance matrix computed around a point. The 50 closest neighboring points were used during data processing. After naming the eigenvalues as $\lambda_1 > \lambda_2 > \lambda_3$, the computed features are:

- Anisotropy: $A_\lambda = (\lambda_1 - \lambda_3)/\lambda_1$;
- Planarity: $A_\lambda = (\lambda_2 - \lambda_3)/\lambda_1$;
- Sphericity: $S_\lambda = \lambda_3/\lambda_1$;
- Linearity: $L_\lambda = (\lambda_1 - \lambda_2)/\lambda_1$;
- Change of curvature: $C_\lambda = \lambda_3/(\lambda_1 + \lambda_2 + \lambda_3)$.

"Local plane-based" (iii) features may be used to discriminate between buildings and vegetation. Once a local plane is robustly estimated using a set of neighboring points as support, the following features can be computed:

- $N_z$: the deviation angle of the normal vector of the fitted plane from the vertical direction;
- $R_i$: residuals of the local estimated plane;
- $\sigma_N^2$: the variance of the point normal concerning the normal vector of the fitted plane.

A training set was manually selected to train RF according to the following classes: "tree," "ground," and "building." An example of RF classification output is shown in Figure 6.

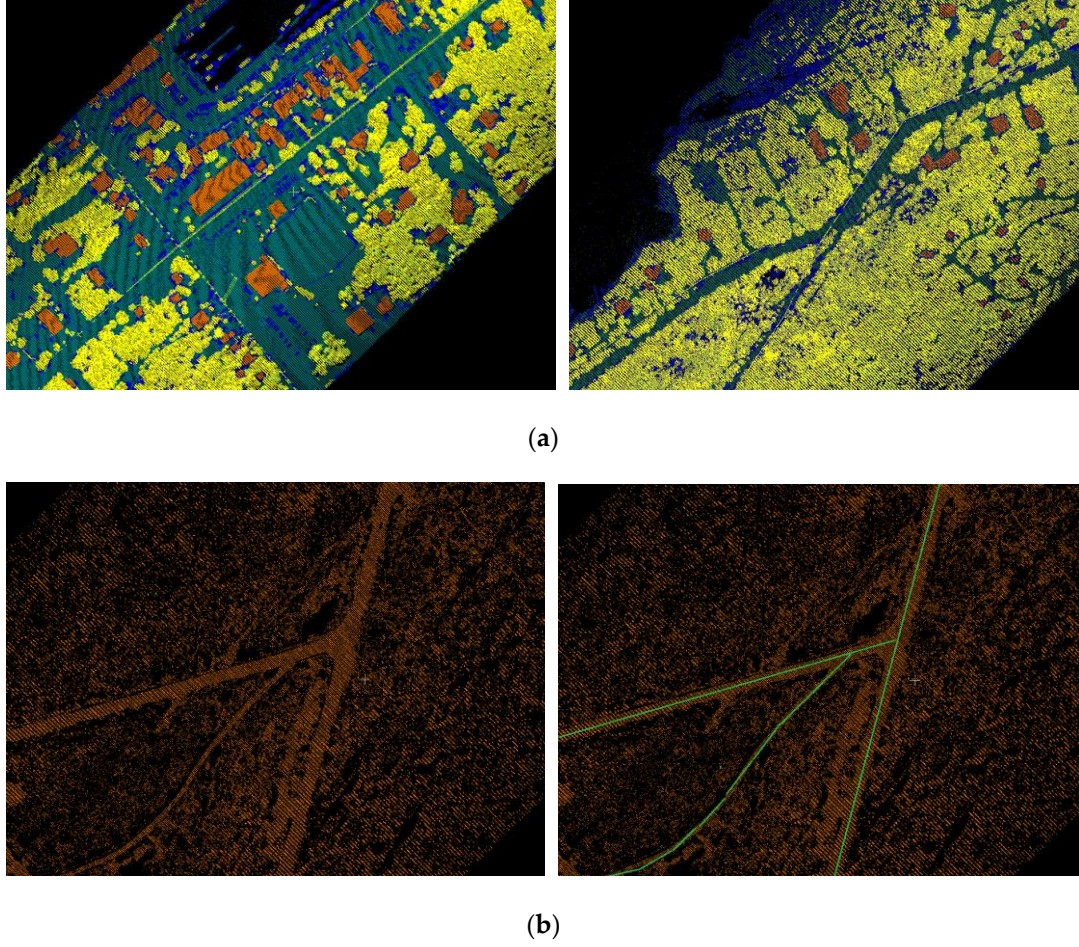

(**a**)

(**b**)

**Figure 6.** Point cloud classification using RF: (**a**) the output for the urban area of Tobermory (**left**) and a rural area (**right**). Points are colorized according to the identified class (building = red, tree = yellow, ground = green, unclassified = blue); (**b**) an example of ground segment in a rural area, roads are characterized by a high point density for bare soil.

*4.2. Road Centerline Detection*

At this stage of the processing, the layer named "ground" contains roads, parking lots, bare ground, low-land grass, etc. However, compared to roads, parking lots and bare soil have different shape features. Points belonging to a road are generally denser than points on bare ground under trees. They also present a linear shape. The first stage for detection of the road centerline is the definition of linearity for point distribution at the local scale. Indeed, roads are characterized by a higher density of points aligned along a line. Bare ground presents a lower density of points compared to roads and the roughness of the surface is generally higher.

The local planarity of roads is investigated at different scales using a multi-level architecture to take into consideration different road width. As previously discussed, a road can be identified in a point cloud as a dense linear segment with a smooth shape. For this reason, a squared area is investigated around each "ground" point (named "seed"). A line is firstly fitted to the local set of points using RANSAC [50]. A tolerance value equal to one-third of the side of the clustering window (with a maximum of 8 m) is considered for linear fitting. A minimum number of support points is also used for line acceptance, which is equal to one-third of the points in the squared sample. To prevent possible over-segmentation, i.e., identification of linear features that do not exist, region growing is applied to all points in the clustering window using as new seed the line detected by RANSAC. The following characteristics are evaluated: point smoothness and point intensities at 532 nm, 1064 nm, and 1550 nm.

If the line is accepted, the local surface smoothness is computed as the average between the elevation $z_j$ of a point $p_j$ and the average elevation of all points on the line. Similarly, the planarity of one of the remaining points $p_i$ in the window is computed considering the average elevation of the 25 nearest neighborhood points of $p_i$.

The final shape of the region is then evaluated by computing the linearity coefficient ($L_\lambda$). If $L_\lambda > 0.5$ the feature is classified as linear and all points belonging are used in a voting system. Otherwise, no votes are assigned to points and the original "seed" point is penalized. Some examples of typical cases of voting are presented in Figure 7.

Once linearity voting is carried out, separation among different point classes can be performed. The specific spatial point distribution associated with each object class reflects different linearity votes. Besides road points, the detected "ground" points contain parking lots, bare ground, and low grass. However, compared to parking lots, bare ground, and low grass, road points feature a higher vote. Therefore, the subsequent process is finalized to recognize road points by votes and to extract the primitives of road centerlines. Mean Shift Clustering (MSC) [51] is applied to separate "road" points from "non-road" points. The following characteristics are evaluated per each point: the number of votes received at each specific scale, LiDAR intensity, and point smoothness. MSC allows for the detection of point groups having similarities in terms of the three parameters. Points characterized by a lower number of votes and lower smoothness at each level can be recognized as "off-ground points "or "bare ground/low grass."

Points characterized by a lower number of votes and high smoothness can be categorized as "parking lot," and points with higher votes and smoothness at least in one level are classified as "road." After voting, points classified as "roads" can be transformed into a polygonal representation by using $\alpha$-shapes. An $\alpha$-shape is defined as a frontier, which is a linear approximation of the original shape [52] to be used for boundary reconstruction from an irregular point cloud.

The parameter $\alpha$ controls the precision of the boundary. In particular, the value of $\alpha$ represents the radius of a rolling circle around the point cloud. The rolling track of the circle forms the boundary of the point-set, which also allows for the regularization of the shape. Some spurious points could also remain in the "road" data set (mainly in road shoulders), estimating the road boundaries quite inaccurately. Starting from the polygon of the road, the centerline can be extracted by considering only the boundary points and by determining the centerline with the methods based on Thiessen polygon [53].

| | | |
|---|---|---|
| a. | | Points in the clustering windows around the "seed" (red point) are not recognized as forming a linear cluster, and no votes are assigned. |
| b. | | Points in the clustering windows around the "seed" (red point) are recognized as forming a linear cluster, and a vote is assigned to them (green points). |
| c. | | Points in the clustering windows around the "seed" (red point) are not recognized as forming a linear cluster, and no votes are assigned. |
| d. | | Points in the clustering windows around the "seed" (red point) are recognized as forming a linear cluster, and a vote is assigned to them (green points). A penalization is assigned to the "seed" since it is not part of the linear cluster. |
| e. | | Points in the clustering windows around the "seed" (red point) are recognized as forming a linear cluster, and a vote is assigned to them (green points). |
| f. | | Points in the clustering windows around the "seed" (red point) are not recognized as forming a linear cluster, and no votes are assigned. |

**Figure 7.** Some typical cases of voting: (**a**,**b**) votes at different scales for a road point, (**c**,**d**) votes for a bare ground point, and (**e**,**f**) votes for a point in parking lot areas.

The output centerlines may be quite jagged because of the noise in the original polygons (Figure 8a,b). Additionally, some road intersection areas can be missing. Indeed, due to the characteristics of the voting scheme, intersection areas tend to be classified as "parking lot" (Figure 8c,d). For this reason, a regularization was carried out to form a complete road network. A hierarchical grouping method was adopted to reconstruct the final road network. First, adjacent primitives with similar alignment (according to a user-defined threshold) were connected into longer road segments. In a second stage, connectivity was established by building a cell complex by connecting and intersecting smooth road lines in a way similar to the work presented in [45]. Finally, the removal of short lines that were likely not meaningful roads was carried out (Figure 8e,f).

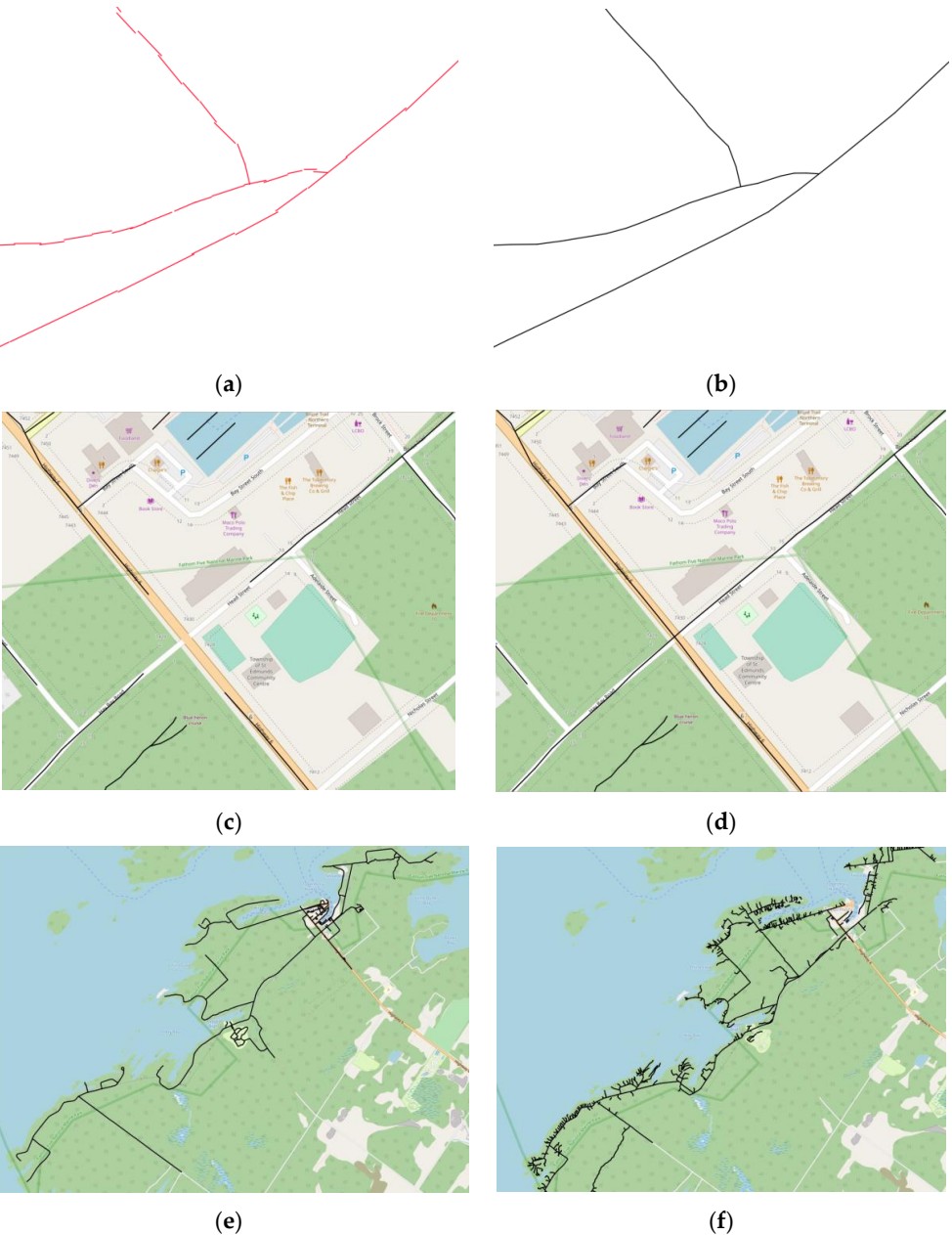

**Figure 8.** Road centerline extraction: (**a**,**b**) example of road line regularization, not regularized (on the left) and regularized (on the right), (**c**,**d**) road connectivity before (on the left) and after (on the right) the cell complex connectivity, and (**e**,**f**) Tobermory road network, OpenStreetMap (OSM—on the left) and the roads extracted with the proposed method (on the right).

## 5. GIS Layers of Roads Turned into BIM Objects Roads

Three-dimensional parametric modeling in the case of roads is quite different than parametric modeling for buildings. Indeed, buildings are assembled using BIM software and the typical architectural elements. BIM for roads requires other typologies of objects for which there is limited availability. These elements are necessary to provide a detailed description of roads and their components, starting from the cartographic level to the fine details of small ancillary accessories, such as traffic lights and signals.

The commercial market offers some software packages for road modeling based on BIM. Examples are Autodesk InfraWorks®, SierraSoft Roads®, Bentley OpenRoads®, Plateia® by CGS Lab, and Line Design Pro® by Trasoft Solutions. The tools available in these software packages consist of solutions for advanced parametric modeling as well as algorithms for simulations.

The software tested in this research work is Autodesk InfraWorks® (AIF), which can handle different types of infrastructures such as roads, bridges, and drainage projects. AIF is a conceptual design software based on operations that are directly performed in a georeferenced 3D space. Detailed infrastructure design requires additional integration of AIF models with other software packages, such as Autodesk Civil 3D® and Autodesk Revit®. In other words, AIF is particularly indicated to experiment with different design proposals and proofs of concept.

The case study analyzed in this paper was modeled using AIF starting from a set of existing data and new geometric information provided by the LiDAR filtering procedure discussed in the previous Section 4. The list of data sources already available for the area includes:

- DTM, provided as raster format (GeoTiff);
- RGB orthophoto, provided as a raster image (GeoTiff);
- Water bodies provided as a vector layer;
- Buildings as a vector layer (from OSM);
- Roads as a vector layer (from OSM);
- LiDAR point cloud.

Additional information has been then generated using the LiDAR point cloud (see Section 4), obtaining vector layers stored for the following class of objects:

- "Road," represented as linear features classified as primary and secondary roads;
- "Vegetation," represented as multiple polygonal layers;
- "Building," represented as multiple polygonal layers.

The mapping reference system of the project is UTM-WGS84, Zone 17N. Data available as geographic coordinates (longitude, latitude) are automatically converted into the chosen mapping grid. Georeferencing is a fundamental aspect of BIM-GIS projects for infrastructures, whereas it is less important for BIM projects at the level of an isolated building. AIF can operate the transformation from geographic to map (cartographic) coordinates after reading the projection file associated with the different data. If the projection file is not available, the reference system can be manually set after importing the layer. Although transformations between different reference systems are available in AIF, it is the authors' opinion that the result should be always validated, especially in the case of those national reference systems for which the parameters available in commercial software could be an approximation of the official parameters.

The BIM-GIS model was setup using different "Proposals." A proposal is the state of the project saved at a specific stage. The project can have multiple Proposals so that users can see the changes that occurred to the model. Proposals are a quite innovative way to store results, especially for users coming from the world of geospatial information, since they do not refer to different project versions stored in multiple files. A single project file contains multiple versions, which can be accessed at any moment without losing information, showing changes, and analyzing different variants. Moreover,

changes in a specific proposal are not reflected in another one, thus, multiple independent projects can be generated in a single environment.

The proposed workflow is based on multiple proposals structured in a specific way. For instance, Proposal 1 is a basic project based on DTM, RGB orthophoto, water bodies, buildings, and roads available from online repositories. Proposal 1 can be generated using the Model Builder® plugin of AIF. The user can specify the area of interest in a geographic viewer and the first version is automatically generated within a few minutes. Moreover, objects are directly related to the DTM, so that they are draped on it. A first inspection revealed that some data downloaded from online repositories were not very accurate. Most buildings have an approximated height and shape, the roof is usually modeled with a flat surface (although in some cases more detailed roofs are also available), and the texture applied to the facades is an artificial reconstruction (Figure 9).

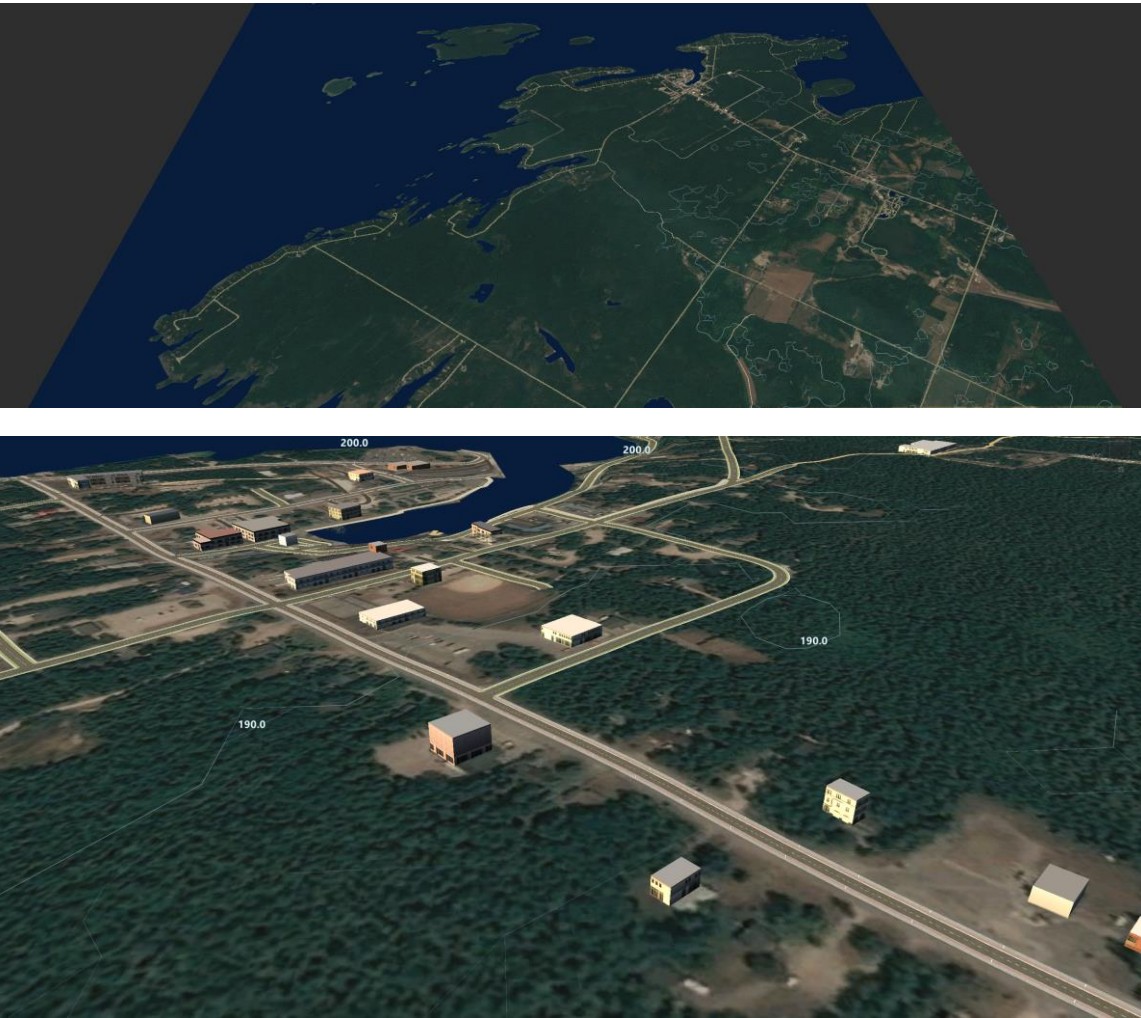

**Figure 9.** Some images of Proposal 1, which was generated using online repositories accessed with Model Builder® in Autodesk InfraWorks® (AIF). Top: an overall view of the whole area. Bottom: an example showing the level of detail for the BIM-GIS model.

The second step for BIM-GIS generation consists of the replacement of the road layers using the vector information generated from LiDAR data, i.e., the creation of Proposal 2. The road vector layer has an extra-attribute that separates "primary" and "secondary" roads. The layer is split into two new layers, which correspond to the different typologies of roads. In the case of multiple road typologies, a new vector layer should be created for each typology.

Primary and secondary roads are then imported with specific road styles, which are standard two-lane roads and a dirt road (or track) in this project. Additional information for roads consists of maximum speed, the number of lanes forward and backward (a road is a parametric object and a modification in the project database is directly reflected in the 3D model), geometric parameters such as the elevation offset or maximum slope, and lifespan (time of creation and dismission). It is also possible to integrate additional information such as special tags or links to some external resources.

Once (two-dimensional) roads are imported with the correct style, the longitudinal profile is automatically draped on the DTM to obtain a 3D representation. Indeed, the used vector layers derived from LiDAR are two-dimensional. The third dimension is given by the height values of the DTM. Problems can arise with bridges, tunnels, or overpasses, for which a complete automatic solution is available in the proposed workflow. If a road intersects a water body (lake, river, etc.), a bridge is automatically added. However, manual corrections after a visual inspection are still necessary and will be discussed during the generation of the next proposal stage.

Figure 10 shows Proposal 1 versus Proposal 2. Roads from OSM have been replaced with the roads detected from LiDAR data, proving a more exhaustive description of the road networks. Results are more complete especially for secondary (dirt) roads, which are not available in OSM.

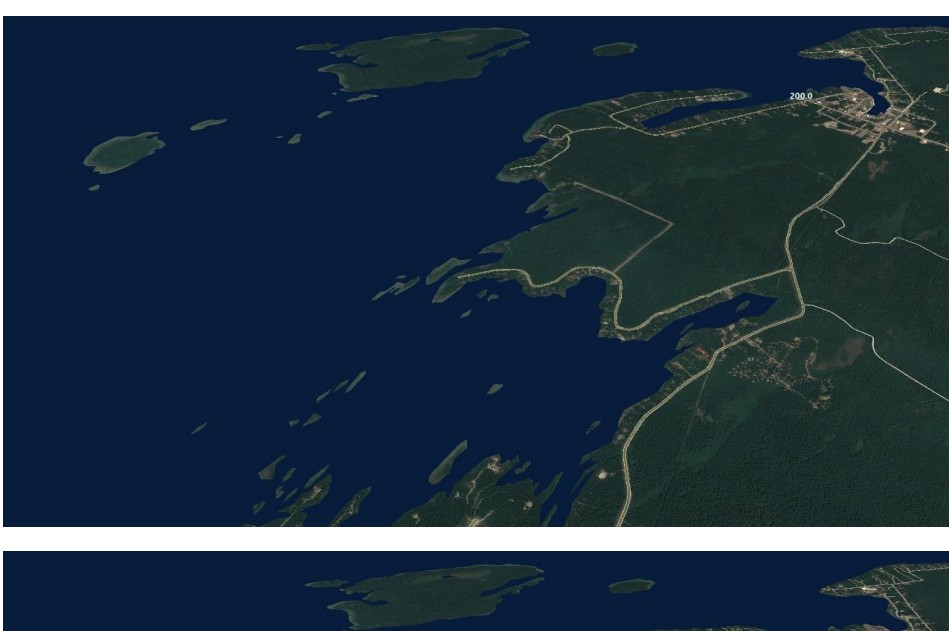

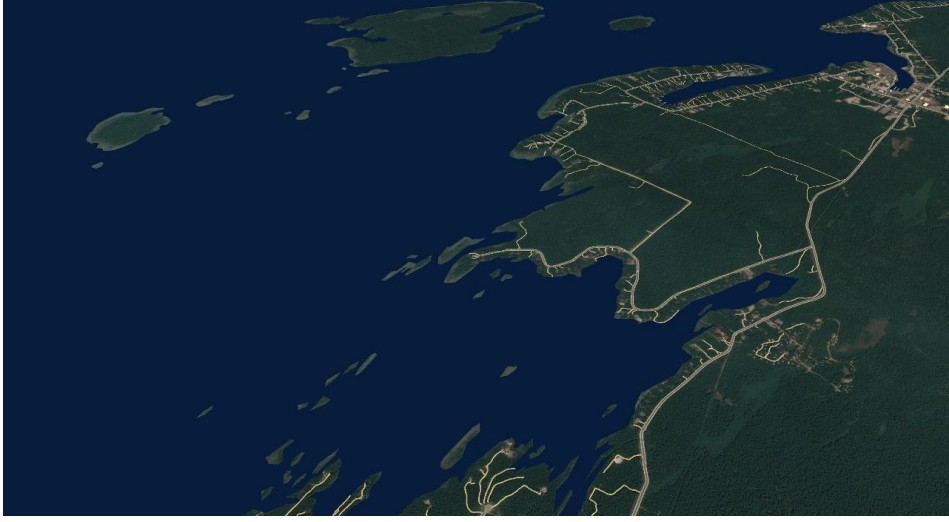

**Figure 10.** On the top: Proposal 1 (roads from OSM); at the bottom: Proposal 2 (road from LiDAR data detected with the implemented classification method).

A simple way to verify the improvement achieved from Proposal 1 to Proposal 2 is a basic change of visualization in the software. Some results are shown in Figure 11, in which Proposal 2 features much more details and a large number of secondary roads. After comparing the point cloud with the results of both proposals, it was also possible to confirm that the roads detected from LiDAR data are metrically more accurate than the ones from OSM.

(Proposal 1)                                     (Proposal 2)

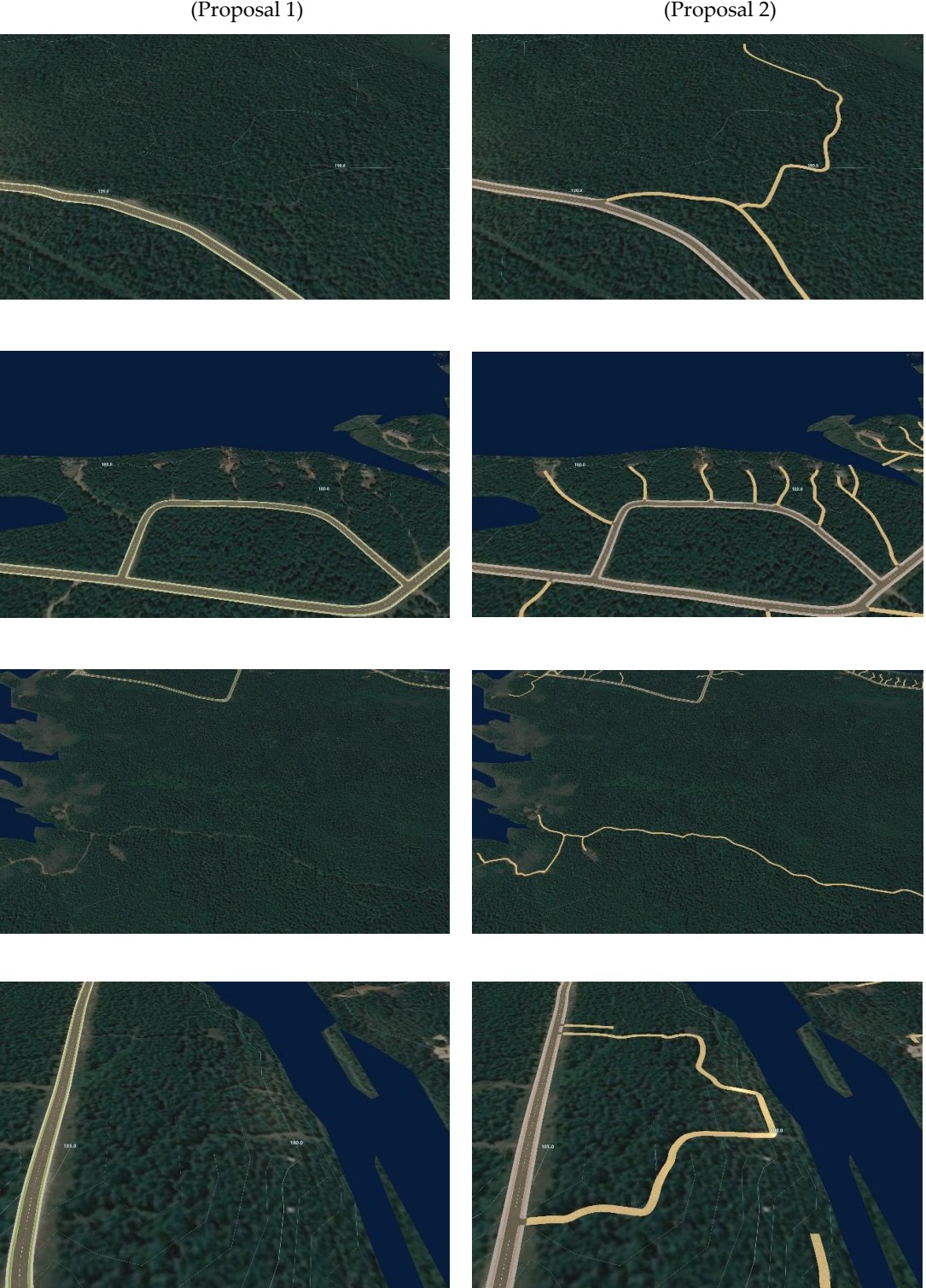

**Figure 11.** Comparison between Proposal 1 and Proposal 2.

The parametric model with the roads generated from LiDAR data also has some errors, which required further editing, mainly consisting of manual corrections. Proposal 3 is intended as an error-free version of Proposal 2.

The current workflow still requires interactive editing carried out by a user who (i) has to inspect the 3D model to identify errors and (ii) apply the corrections. Considerations about the time and the overall effort for manual corrections deserve to be discussed. As things stand at present, no automatic solution is available for validating the integrity of the road in the BIM-GIS. Future research work is required for developing algorithms able to identify and correct errors in the conversion of vector layers into parametric road objects. Such algorithms should inspect the continuity of road, the angles when the road splits or intersects other roads, the presence of bridges, tunnels, and overpasses, the generation of small road segments that are not connected to other roads, etc. As mentioned, such corrections are manually carried out for the lack of algorithms able to deal with these problems.

The user performs a visual inspection and applies local refinements to eliminate geometric inconsistencies. Although the identification of error is still a completely manual operation, the correction can rely on the parametric capability of BIM objects. For instance, when two elements are not properly connected, the user has only to refine the position of a road segment by interactively dragging one segment onto the other. The software will recognize the overlap and join the disconnected segments into a single object. Since objects have parametric intelligence, errors can be easily fixed. On the other hand, the identification is not automatic, and this makes the overall workflow not automatic. Future work in automatic error detection could be carried out exporting the objects in BIM-GIS with conversion in GIS features (e.g., road objects can be turned into polygonal shapefiles). The analysis of overlapping areas could be carried out in GIS software using tools based on the calculation of intersecting polygons. On the other hand, the implementation of procedures for automatic detection is out of the scope of the presented work, as previously discussed.

It is not easy to define the manual effort required to remove errors in different projects. The used dataset is relatively small, and errors were identified and solved in a few hours. The parametric capabilities of BIM software are very useful to correct the mistakes since it is sufficient to (manually) readjust the position of elements so that they can be identified as corresponding elements.

The authors are aware that the proposed dataset is rather small compared to the huge amount of information that can be used in real projects. More research work should be carried out for detecting possible errors in the model, in a way similar to what happens in BIM projects for buildings using methods for interference detection, which, however, do not provide a complete solution also in those projects.

The next figures show some of the errors and the procedure used for their removal. The first problem concerns intersections. Although an intersection between primary and secondary roads can be automatically generated, the very narrow angle between intersecting roads and may cause errors during the use of automatic procedures. In other words, when roads intersect at an angle close to a right angle, the automatic generation of intersections provide good results. On the contrary, when the intersection is more like an acceleration/deceleration lane, manual correction is mandatory. The problem becomes also more significant when two roads with a different road style intersect.

An example is shown in Figure 12, in which the user had to interactively drag the vertices of two roads to form an overlapping area. The software was then able to generate a suitable intersection. The result is a deceleration lane obtained with a semi-automatic approach. The advantage in using parametric objects is mainly related to the opportunity to remodel without redrawing, such as in the case of software for "pure" geometric modeling (e.g., Rhinoceros®, Autodesk Maya®, AutoCAD®) where elements must be re-traced.

(Proposal 2)

(Proposal 3)

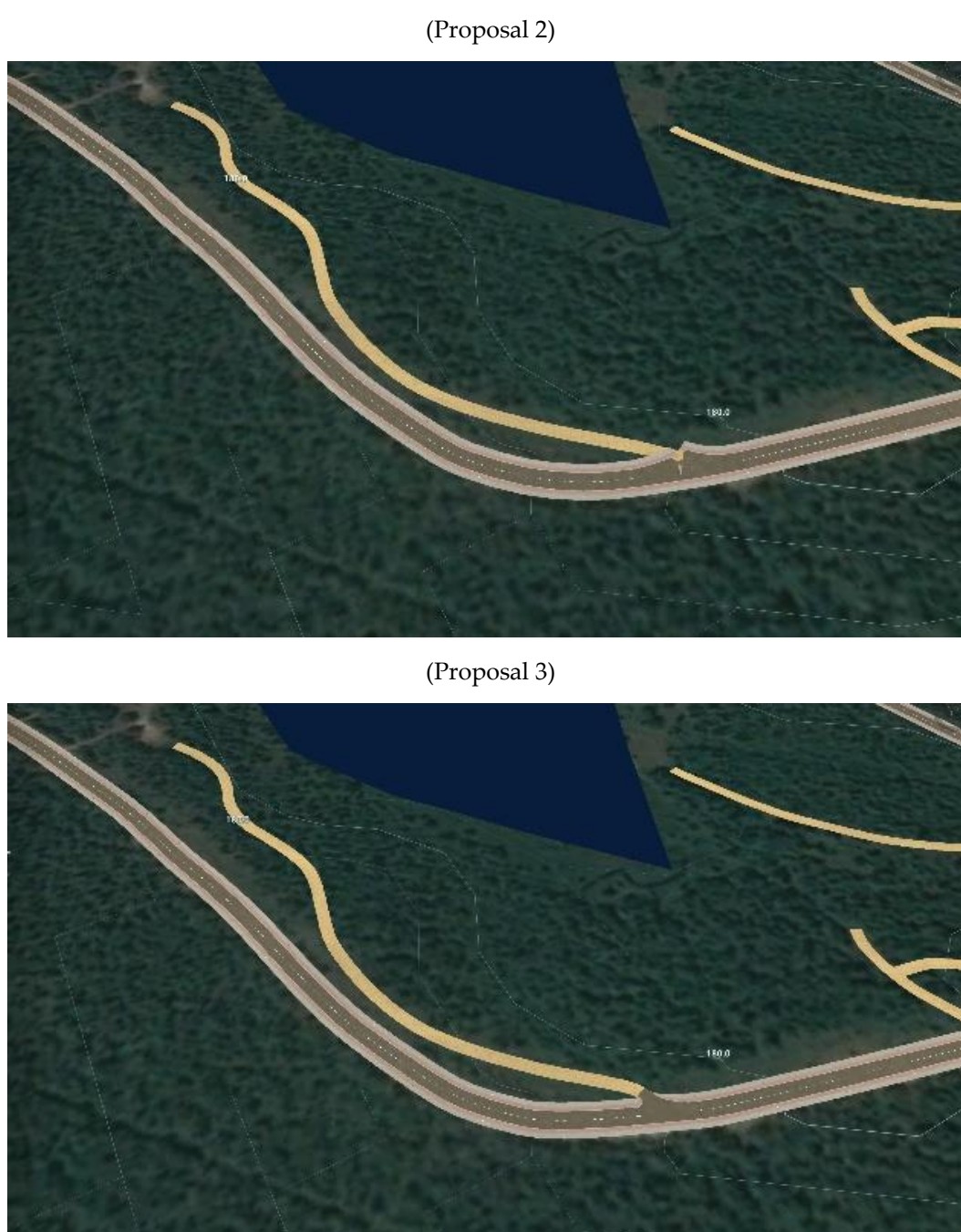

**Figure 12.** Correction of a wrong intersection caused by the small intersecting angle between two roads.

Another error that required manual editing is the sudden interruption of roads, resulting in an incomplete reconstruction. As the algorithm for road extraction from LiDAR data seeks for linear features, a sudden variation of the geometry could lead to the failure of the automatic detection. An example is shown in Figure 13, where a large area along the road with a local lack of vegetation has been misclassified. The correction of this error was carried out by (manually) drawing an additional dirt road segment that connects the final points of the detected road segments. Updated orthophotos and the point cloud may provide very useful information to inspect and revise the model. As can be easily understood, the manual correction of such typologies of errors is relatively simple. The identification could be more problematic, especially for large datasets including different types of roads.

(Proposal 2)

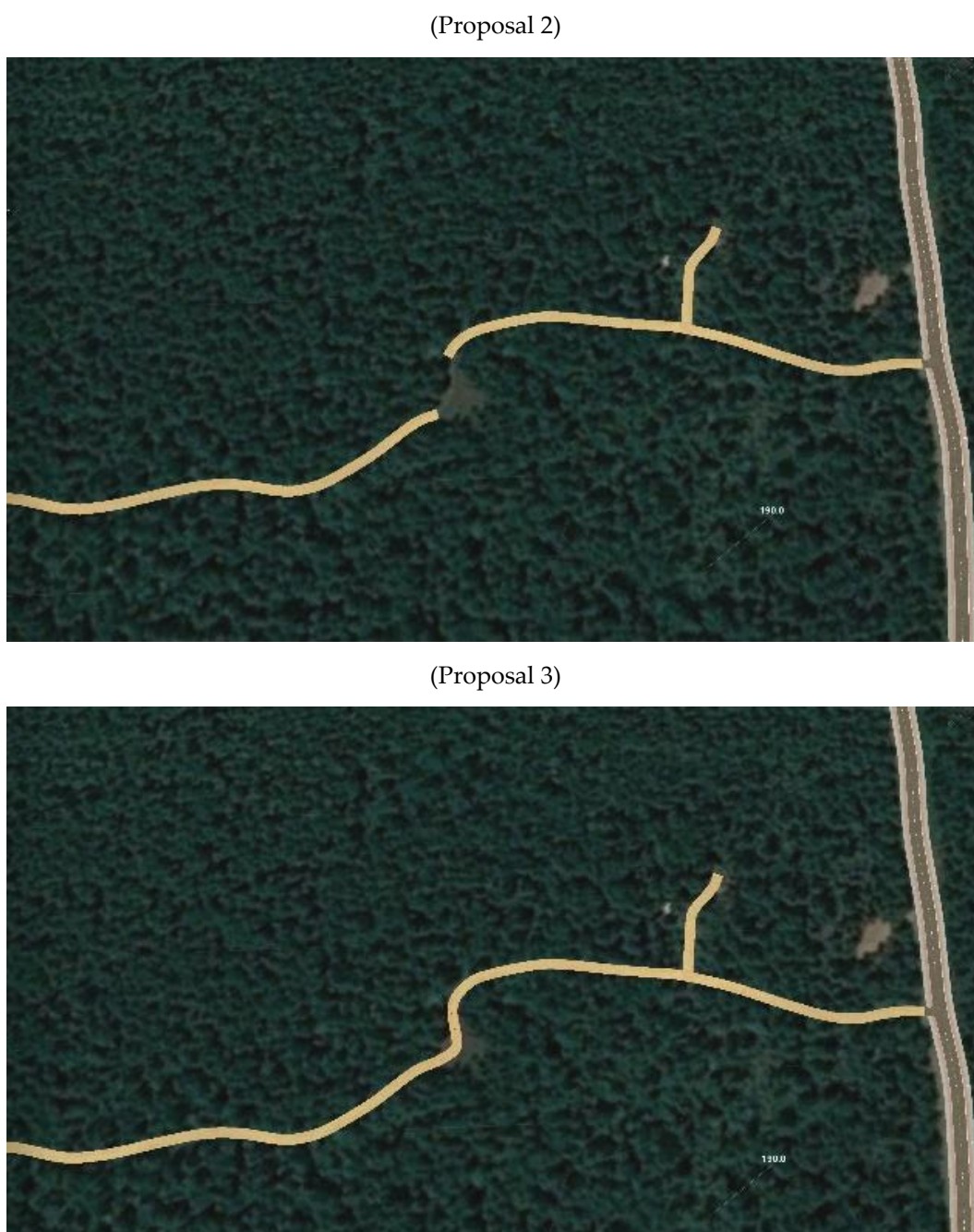

(Proposal 3)

**Figure 13.** Correction of a sudden interruption of the road caused by an enlarged area with less vegetation.

Finally, the last type of error detected consists in the wrong classification of roads. Figure 14 shows a primary road with two internal segments classified as dirt roads. As can be seen, parametric modeling was able to fix the transition between the different types of roads, notwithstanding the initial classification. On the other hand, this gives a wrong result that could cause several issues in future operations, such as traffic simulation in which numerical analyses depend on the characteristics (such as speed, number of lanes, etc.) associated with each road style.

The correction is a simple operation. The operator has to choose the misclassified road segments and change the road style in the database. The connection with other roads is then automatically handled.

(Proposal 2)

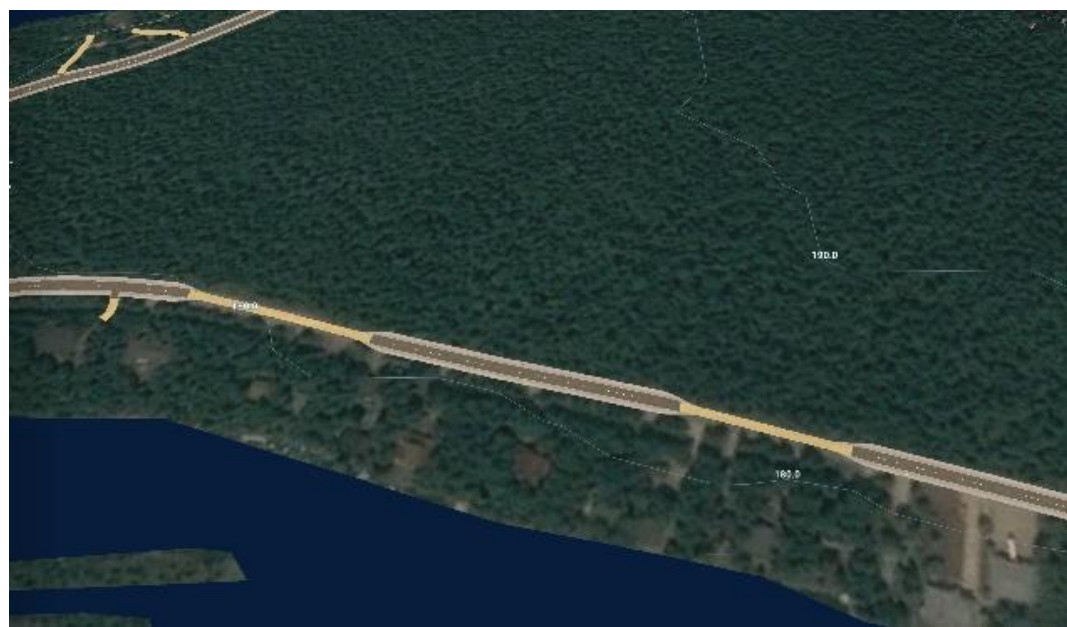

(Proposal 3)

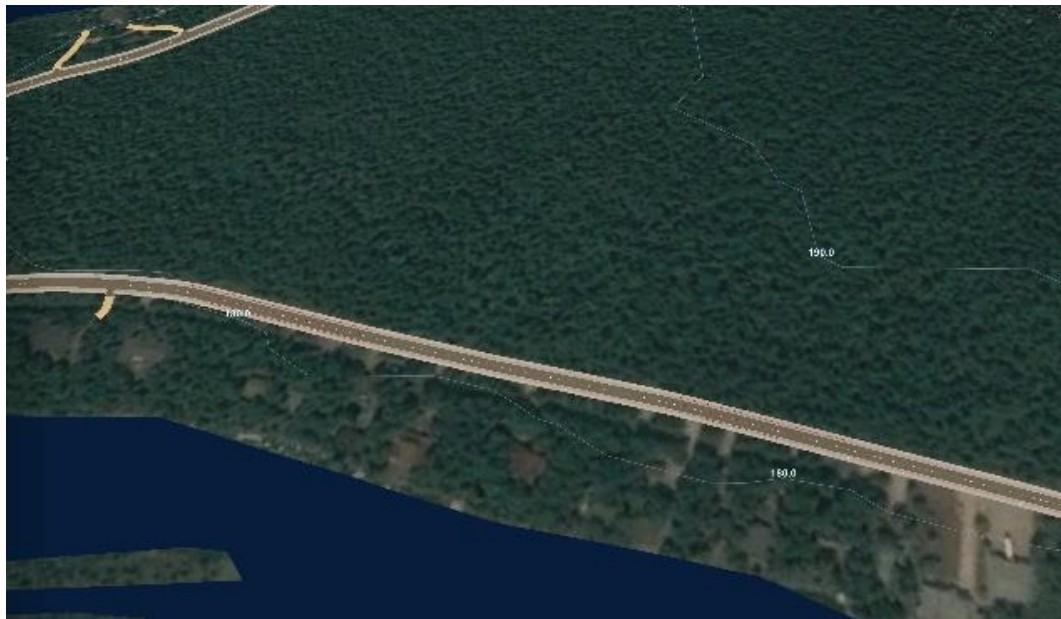

**Figure 14.** Problems with misclassified roads: the software was able to handle the transition between different road typologies, resulting in an unrealistic model for the road.

Thanks to the parametric modeling capabilities of AIF, the correction of the detected problems is not a complex operation. What is less simple is the identification of all errors, for which a manual inspection could not only require a long time (a few hours for the relatively small dataset used) but also result in errors not correctly detected, especially in the case of large datasets. As previously mentioned, the need for automatic algorithms able to check the result after the conversion of GIS layers into parametric BIM objects is a fundamental requirement for effective real applications. Probably, these corrections should be carried out not only in the BIM-GIS, but also in the GIS environment before importing data in AIF, or converting and exporting the BIM-GIS layers into a suitable GIS format,

in which additional checks can be carried out. Some issues, such as those shown in Figure 12 (wrong generation of intersections with a small angle), are instead mainly related to parametric modeling and require a correction in the BIM-GIS environment. At present, the only possible way to revise such mistakes is manual editing with the support of parametric modeling tools, orthophotos, and the LiDAR point cloud. Future research work will be necessary to address such technical challenges.

## 6. Strategies for Model Enhancement: Buildings and Vegetation

The procedure for the automatic classification of LiDAR data described in previous sections may also provide the geometric information about vegetation and buildings. These new layers can be incorporated into the BIM-GIS model and used for further analyses, such as traffic flow simulation, in which buildings and vegetation may limit driver visibility.

Proposal 4 consists of the model with vegetation, which is added as a parametric element. The level of detail for vegetation layers in BIM-GIS software starts from a single element (e.g., a single tree whose position is given by a point). Trees can also be organized along lines (rows of trees) or surfaces (stands of trees), in which a surface can be covered by trees with homogenous or variable spacing. In this last case, a polygonal vector layer can be used to represent the surface to be covered with trees. The parameters to be set in the software database consists of the typology of the tree, its three-dimensional size (such as a variable scale factor in three directions), height, density between different trees, lifespan, etc.

An example of the final model (Proposal 4) is shown in Figure 15. As can be seen, vegetation is not distributed over the entire area but only on those parts identified from LiDAR data. The reconstruction does not reveal the real position of independent trees. The main focus of this paper is the reconstruction of roads. Hence, the areas covered by trees are detected with a random distribution of vegetation, which is quite dense for the considered area. The global number of trees is also available and can be used to compute the volume occupied by vegetation, which can be compared to the result achieved from LiDAR. In such figures, the same type of plants was used for the entire area. On the other hand, multiple species can be simultaneously added in the same area.

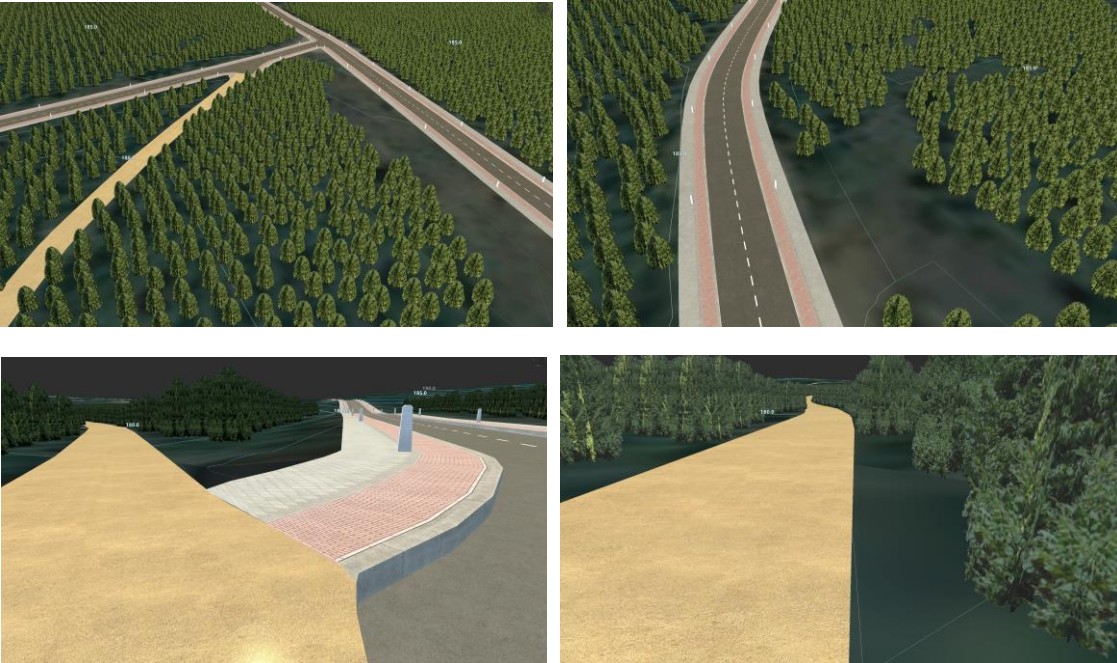

**Figure 15.** Some images of vegetation detected from LiDAR data classification added as a parametric object in AIF.

After modeling vegetation, the last layer derived from LiDAR concerns buildings. The classification procedure may identify points measured on buildings. A vector layer with the boundary of each building is automatically generated and imported in AIF. The low spatial density of the adopted LiDAR data set provided the boundary of the detected building, as can be seen in Figure 16. A regularization algorithm was applied directly to the polygons detected to obtain a more rectangular-like shape.

The regularization works iteratively (Figure 17). In the first step, original polygons are substituted by their bounding box. The area of the original polygon is then compared with the area of the bounding box using a threshold (5% of the original area). This allows regularizing buildings that have a rectangular-like shape. If the planar projection is more complex, i.e., more rectangles connected, the regularization requires additional iterations. If the difference between the area of the oriented bounding box and the original polygon is larger than 5%, an iterative subtraction procedure can be applied. The difference between the bounding box and the original polygon is computed to identify 'holes.' Large 'holes' are substituted by their bounding boxes. Then, the new polygons are iteratively subtracted using bounding boxes.

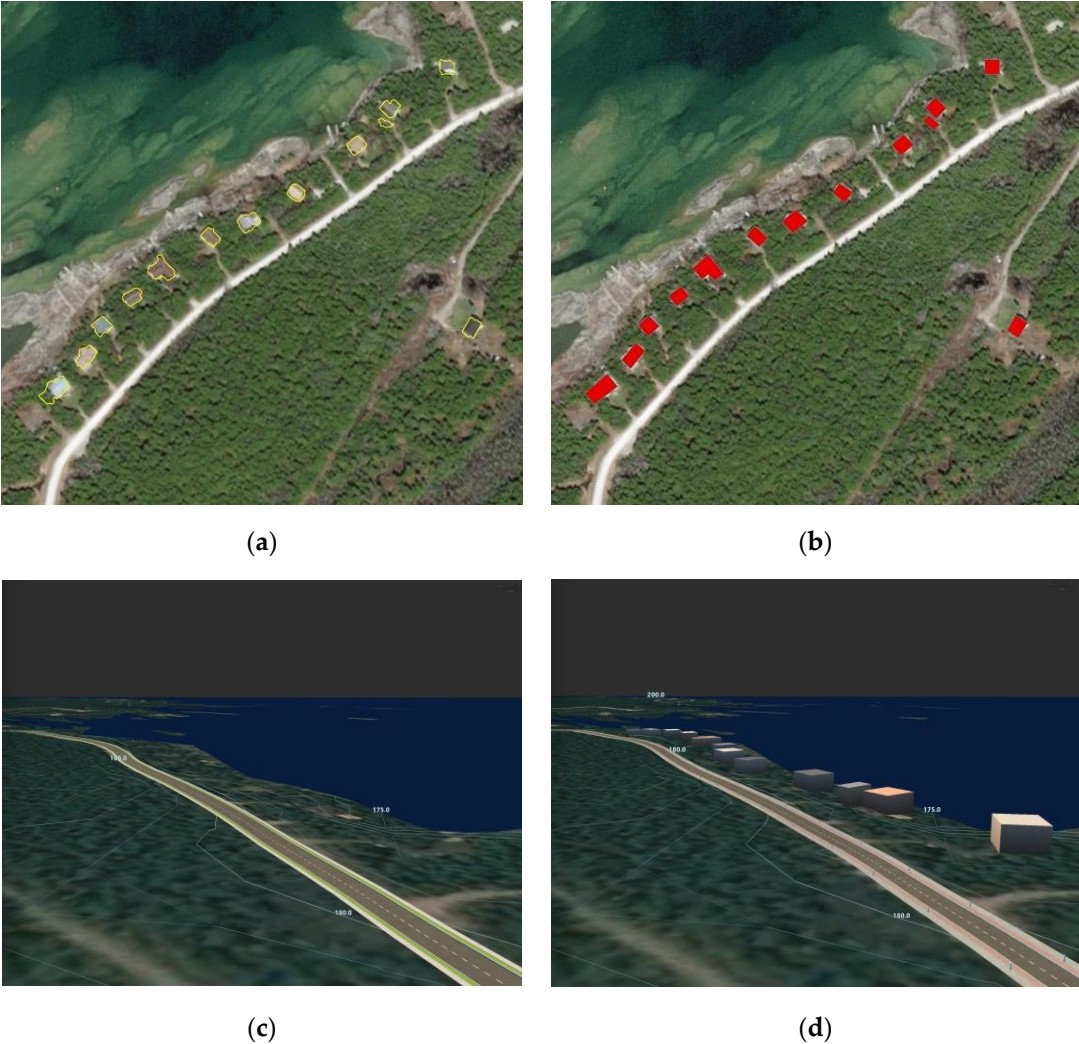

(**a**)          (**b**)

(**c**)          (**d**)

**Figure 16.** Simplified building models (**b**) obtained in AIF from building footprints (**a**) detected from LiDAR data classification. At the bottom: the model from OSM, where buildings are not available (**c**) and the result achieved with the proposed approach (**d**).

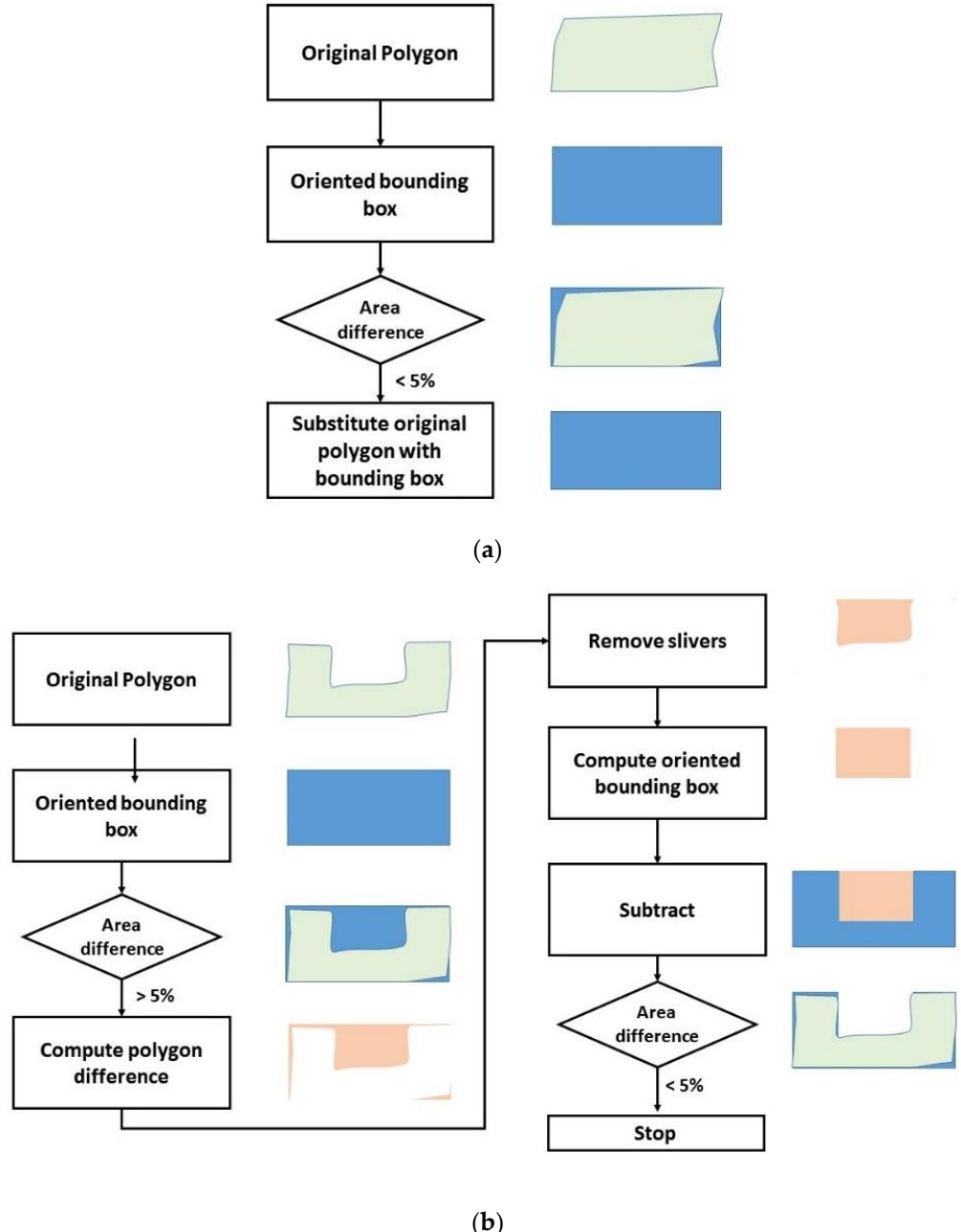

(**a**)

(**b**)

**Figure 17.** Building regularization example: the case of a rectangular-like building (**a**) and the case of a more complex building plan (**b**).

The vector layer can be added to AIF project to obtain Proposal 5. The vector layer is also draped on the DTM and building height is added using a simplified representation based on vertical extrusion (flat roof). The average height of each building can be obtained from the point cloud and is stored inside the vector layer. The BIM-GIS system can then use this attribute to generate a solid building model.

Additional elements that could be added to this model are the small road accessories (traffic lights, signs, barriers, etc.). On the other hand, the density of the point cloud does not allow the reconstruction of such small details, which would require a manual modeling approach because the proposed method is not able to detect small elements. Another output available from LiDAR data filtering is DTM. The interpolation of points classified as "ground" provides a raster grid with elevation values. The analysis of the data quality of the obtainable DTM is out of the scope of this paper and will be investigated in the future.

## 7. Conclusions

Parametric 3D modeling based on BIM at the scale of infrastructures is becoming more important in a huge variety of applications, opening the use of BIM outside the architecture and construction industry. The reuse of existing geospatial information (orthophotos, digital terrain models, vector layers, etc.) integrated with other data such as LiDAR point clouds is an essential tool for the generation of accurate and reliable BIM-GIS models of infrastructures.

This paper has presented a workflow for LiDAR data classification in which different categories of elements are extracted and turned into vector layers, and imported and combined with existing geographic information to generate 3D BIM-GIS models. LiDAR (or more in general point clouds) have become a standard in the production of BIM at the level of the building (using terrestrial laser scanning). Nowadays, some producers of commercial BIM software have already integrated some tools to handle point clouds in their packages.

The proposed methodology for road centerline extraction can handle moderate change in point cloud density (15–20%) into each strip. In the case of larger changes, a further pre-processing step can be included to subdivide the point cloud into homogenous areas. Extension of the proposed method to point clouds acquired with terrestrial mapping vehicles will require specific adaptations since problems related to variable point density are more significant. However, the proposed multi-scale approach and the proposed voting scheme can be adapted considering different buffer zones around the scanner, each of them characterized by different density values.

Point clouds are mainly used as visual support for advanced modeling operations. Some software packages also include advanced tools for filtering, classification, and 3D modeling, notwithstanding that a manual check and correction is still a mandatory part of the processing workflow. The creation of the integrated BIM-GIS has revealed that manual correction of errors during the conversion of vector shapefiles into parametric road objects is an important issue that deserves future research work. No automatic system for error detection and correction is available and only the interactive inspection by (human) operators can identify errors. Possible solutions could be the re-conversion of the road object into GIS layers and the comparison with the original one. On the other hand, such aspects need to be investigated in future work.

Although the proposed procedure integrates an algorithm for road extraction from LiDAR, the generation of the BIM-GIS environment can be independent of the methods used to produce input vector- and raster files. This allows the reuse of existing cartographic layers, as well as the generation of road profiles with other algorithms and procedures. Such flexibility is important to facilitate the creation of integrated BIM-GIS environments in several applications involving multiple specialists.

As the proposed workflow uses commercial software as output environments, the need to integrate the output with current BIM-GIS standards is an important matter. The interoperability of different BIM-GIS platforms should also consider the rapid development in the IFC standards for infrastructure, which has received more attention in the latest years. IFC 5 is expected to introduce relevant novelties in multiple infrastructure domains (e.g., IFC Road, IFC Bridge, IFC Tunnel, IFC Rail, etc.).

Overall, the paper tried to illustrate and discuss a complete process, which focused on the extraction of parametric roads progressively refined by adding information about the road typology. Classification of LiDAR data has provided accurate results for road reconstruction, which, however, still requires manual corrections, such as in the case of complex road intersections with angles much smaller than 90°. Other cases which required revisions and correction based on manual measurements were discussed, demonstrating that a completely automatic approach for the generation of BIM-GIS model is still not feasible.

Finally, this paper has discussed the integration of additional elements derived from LiDAR data classification. This is the case of vegetation or buildings that are modeled as simplified elements. Future research work is required to obtain reliable BIM-GIS elements not only for such elements but also for water bodies (lakes, rivers, etc.), green areas, parking areas, among others. In other words,

future work is necessary to consider for all those elements traditionally reconstructed in geospatial databases, which could become parametric BIM-GIS objects.

**Author Contributions:** L.B. developed the method to import and parametrize GIS layers into integrated BIM/GIS environments. M.P. implemented the strategy for LiDAR processing and road extraction. M.S. coordinate the activities and supervised the entire process. All authors have read and agreed to the published version of the manuscript.

**Funding:** This research received no external funding.

**Conflicts of Interest:** The authors declare no conflict of interest.

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
