# Peer review of "Roads Detection and Parametrization in Integrated BIM-GIS Using LiDAR"

_infrastructures, doi:10.3390/infrastructures5070055_

Round 1

Reviewer 1 Report

This paper presents a process that uses aerial point cloud data to provide GIS layers that can be integrated in a BIM-GIS model, using existing commercial software such as Autodesk InfraWorks® to create parametric models. The usage of point cloud data plays an important role in the assistance of the GIS-BIM model generation, as automatic procedures are developed for classification (road, vegetation and buildings), and road centerline extraction.

The topic of this paper has a great interest, as BIM projects in linear infrastructure are expected to be more common and relevant in the next few years, especially with the upcoming IFC5 and the already existing IFCRoad as a candidate standard.

The paper is well written and easy to follow and poses an interesting line of future research. There are a few remarks that must be considered before publication:

  • Overall, the technical contribution of this manuscript is arguably low in terms of point cloud processing. It is limited to a point cloud classification using Random Forests and a road centerline extraction using a combination of already existing methods such as RANSAC and MSC. The features employed for RF training are also common in the literature. In order to propose this methodology as a meaningful contribution to the field, it should be compared or validated against other state-of-the-art methods. The reader should learn, at least, why the proposed methodology has an advantage over other existing methods.
  • The conceptual contribution of this manuscript (that is, moving from traditional GIS layers to a parametric 3D modeling based on BIM) is indeed interesting and meaningful. The impact of the point cloud support is made clear to the reader across the different modeling proposals. However, there are several steps involving manual corrections. In P19, L567, authors say “The correction of such errors is also quite simple and can be done in a very short time”. This kind of statement should always come together with validation metrics. What is a very short time? How long does it take to a manual operator to correct the errors that still exist using your point cloud processing methodology? How faster it is with respect to a modeling approach that does not include the proposed automatic processes? Knowing operation and processing times should be relevant for any company interested in BIM infrastructure modeling.

Other minor issues that should be remarked:

- The definition of different output levels (proposals) seems to be the key output of the manuscript. It could be indicated in the abstract to give a better insight on the results of the manuscript.

- P2, L62. A review of BIM for infrastructures is referenced, this can be more complete if more recent reviews are added, such as [1,2].

- This paper uses aerial point cloud data as input. Point density on aerial scans depends not only on the equipment, but on other factors such as flight planning. How would this methodology handle any change on point cloud density? Furthermore, terrestrial mobile systems are frequently used for linear infrastructure analysis since the road is surveyed with the highest resolution. How adaptable is the methodology to terrestrial point clouds?

- P21, L642 “a very irregular shape for each detected building, as can be seen in Figure 16”. Even if the statement is easy to understand, it does not seem that the buildings in the figure have a ‘very irregular shape’. Maybe a zoomed version of Fig16a helps to gain coherence between the statement and the figure.

- P22, L643-655. Fig 17 explains very well the process that is described along these lines. The written explanation could be simpler as it is well supported by the workflow.

In summary, this manuscript presents a relevant approach that is in line with the interest of infrastructure companies of implementing BIM on linear infrastructure projects. Once the key issues remarked in the review have been considered, this manuscript could be proposed for publication.

  1. Costin, A.; Adibfar, A.; Hu, H.; Chen, S.S. Building Information Modeling (BIM) for transportation infrastructure – Literature review, applications, challenges, and recommendations. Autom. Constr. 2018, doi:10.1016/j.autcon.2018.07.001.
  2. Liu, X.; Wang, X.; Wright, G.; Cheng, J.C.P.; Li, X.; Liu, R. A state-of-the-art review on the integration of Building Information Modeling (BIM) and Geographic Information System (GIS). ISPRS Int. J. Geo-Information 2017.

Author Response

We want to thank Reviewer 1 for her/his valuable comments and suggestions. We took all of them into consideration in the revised version of the manuscript. Such corrections have improved the quality of the paper. We report in this document the original comments and our answers.

Comments and Suggestions for Authors

This paper presents a process that uses aerial point cloud data to provide GIS layers that can be integrated in a BIM-GIS model, using existing commercial software such as Autodesk InfraWorks® to create parametric models. The usage of point cloud data plays an important role in the assistance of the GIS-BIM model generation, as automatic procedures are developed for classification (road, vegetation and buildings), and road centerline extraction.

The topic of this paper has a great interest, as BIM projects in linear infrastructure are expected to be more common and relevant in the next few years, especially with the upcoming IFC5 and the already existing IFCRoad as a candidate standard.

The paper is well written and easy to follow and poses an interesting line of future research. There are a few remarks that must be considered before publication:

Overall, the technical contribution of this manuscript is arguably low in terms of point cloud processing. It is limited to a point cloud classification using Random Forests and a road centerline extraction using a combination of already existing methods such as RANSAC and MSC. The features employed for RF training are also common in the literature. In order to propose this methodology as a meaningful contribution to the field, it should be compared or validated against other state-of-the-art methods. The reader should learn, at least, why the proposed methodology has an advantage over other existing methods.

We totally agree with this comment. It is also our opinion that several methods reported in technical literature can be exploited to produce the input for the generation of the integrated BIM-GIS. This would also make the creation of a BIM-GIS more flexible, without restricting the creation of the input files to the proposed method. We clarified such aspects in the revised manuscript. 

We also agree with the reviewer when she/he points out that  the presented methodology for point cloud classification based on Random Forest is quite common in the literature. Defining a new point cloud classification workflow is out of the scopes of the present work. The aim of the proposed paper is to develop a procedure for road extraction starting from a classified point cloud. We tried to improve the description of this concept in the text. In this specific work we used a Random Forest classification workflow. However, any other classification workflow providing similar results can replace the one presented in the paper. This consideration was added in the manuscript. To the best of our knowledge commercial software solutions for extraction of road centreline starting from Airborne Laser Scanning data require quite significant manual work and extensive user interaction. Works dealing with automated road centreline extraction are mainly research works. Compared to previous researches on road centreline extraction we believe that main contributions of our work can be summarized as:

  • The implementation of a multi-scale approach taking advantages of point cloud information at local and global scale allowing the identification of roads (both main roads and dirty roads) characterized by significantly different widths; 
  • The development of a voting scheme allowing to identify and discriminate between roads and other surfaces (parking lots, squares, etc.) characterized by similar local features (local planarity, LiDAR intensity, etc.) in the point cloud; and      
  • The implementation of a regularization procedure allowing for the generation of a road network presenting consistent topological relationship extending to centreline extraction the procedure based on cell complex presented in (Previtali, M., Díaz-Vilariño, L., Scaioni, M. Indoor building reconstruction from occluded point clouds using graph-cut and ray-tracing. Applied Sciences, 8(9), 1529, 2018).

In addition, the paper presents the development of a new procedure for building footprint regularization. We added this discussion in Section 4 “Generation of road GIS layers from LiDAR data”.

The conceptual contribution of this manuscript (that is, moving from traditional GIS layers to a parametric 3D modeling based on BIM) is indeed interesting and meaningful. The impact of the point cloud support is made clear to the reader across the different modeling proposals. However, there are several steps involving manual corrections. In P19, L567, authors say “The correction of such errors is also quite simple and can be done in a very short time”. This kind of statement should always come together with validation metrics. What is a very short time? How long does it take to a manual operator to correct the errors that still exist using your point cloud processing methodology? How faster it is with respect to a modeling approach that does not include the proposed automatic processes? Knowing operation and processing times should be relevant for any company interested in BIM infrastructure modeling.

This is another relevant comment. We agree that the provided description was not sufficiently clear. We could say that defining manual corrections are “simple” is rather superficial. Consequently, we completely revise this part of the text. We added a better explanation to consider the problem of manual correction as an effective challenge, instead of saying that they can be easily corrected with manual measurements. We believe automation in error detection has to be considered as an effective challenge for future research. We mentioned the solution available in BIM software for buildings, such as interference detection. A similar approach could be extended in the case of infrastructure elements. Then, we tried to clarify that the problem is related to two aspects: (1) identification of an error and (2) correction. Step 1 is completely manual, at the moment we just perform a visual inspection, and we clarified that this is a significant limitation for “real” projects in large areas. Step 2 can be partially automated using the parametric capability of BIM-GIS elements. We think that the description in the revised text is clearer and more concise now.

Other minor issues that should be remarked:

- The definition of different output levels (proposals) seems to be the key output of the manuscript. It could be indicated in the abstract to give a better insight on the results of the manuscript.

Yes, we added this in the abstract, and we have added more details on this point in the manuscript since it is something different (and innovative): a single file with multiple project versions instead of multiple files. It reduces also the risk of having inconsistent project version

- P2, L62. A review of BIM for infrastructures is referenced, this can be more complete if more recent reviews are added, such as [1,2].

Thank you very much for this suggestion. We added both papers.

- This paper uses aerial point cloud data as input. Point density on aerial scans depends not only on the equipment, but on other factors such as flight planning. How would this methodology handle any change on point cloud density? Furthermore, terrestrial mobile systems are frequently used for linear infrastructure analysis since the road is surveyed with the highest resolution. How adaptable is the methodology to terrestrial point clouds?

The presented methodology is considering approximately uniform point cloud density for the surveyed area, as in the case of ISPRS dataset. As highlighted by the reviewer point cloud density can vary along a strip or among different strips on the same area. The proposed methodology can handle moderate change in point cloud density (15-20%). In the case of larger changes in point cloud density the proposed methodology can be modified subdividing the point cloud into areas characterized by similar point cloud density and process them separately up to road centretreline definition with Thyssen polygons. The different road centreline can be merged for the final regularization step. For subdividing the point cloud into point cloud “tiles” uniform density a preprocessing step is requested. It can be based on calculation of the point cloud density and subdivision into “tiles” with MSC. 

The extension of the proposed method to MLS needs for some specific adaptations since in this case the problem of varying point density is more severe. However, the proposed multi-scale approach for detecting different typology of roads (both main roads and dirty roads) and the proposed voting scheme can be adapted also in the case of MLS considering different buffer zones around the MLS characterized by different densities and by merging the different buffer zones defining into a unique voting by defining some specific weights. 

Such considerations were added in the paper. 

- P21, L642 “a very irregular shape for each detected building, as can be seen in Figure 16”. Even if the statement is easy to understand, it does not seem that the buildings in the figure have a ‘very irregular shape’. Maybe a zoomed version of Fig16a helps to gain coherence between the statement and the figure.

Our description was quite confusing, indeed the building does not have a very irregular shape, We revised the text, pointing out that the importance of regularization algorithms on the automatically extracted polygons just for the need to get straight lines instead of “undulated” lines that follow the point cloud 

- P22, L643-655. Fig 17 explains very well the process that is described along these lines. The written explanation could be simpler as it is well supported by the workflow.

We agree. We simplified the description in the text.

In summary, this manuscript presents a relevant approach that is in line with the interest of infrastructure companies of implementing BIM on linear infrastructure projects. Once the key issues remarked in the review have been considered, this manuscript could be proposed for publication.

Costin, A.; Adibfar, A.; Hu, H.; Chen, S.S. Building Information Modeling (BIM) for transportation infrastructure – Literature review, applications, challenges, and recommendations. Autom. Constr. 2018, doi:10.1016/j.autcon.2018.07.001.

Liu, X.; Wang, X.; Wright, G.; Cheng, J.C.P.; Li, X.; Liu, R. A state-of-the-art review on the integration of Building Information Modeling (BIM) and Geographic Information System (GIS). ISPRS Int. J. Geo-Information 2017.

This papers have been included in the reference list.

Reviewer 2 Report

Excellent research and clear writing presentation. The topic is actual both in research and practice, and the paper demonstrates the authors' deep knowledge in main technical challenges of BIM, GIS and 3D pointclouds modelling and processing. The comparisons between the main software tools are very interesting and well analysed.    

Author Response

Excellent research and clear writing presentation. The topic is actual both in research and practice, and the paper demonstrates the authors' deep knowledge in main technical challenges of BIM, GIS and 3D pointclouds modelling and processing. The comparisons between the main software tools are very interesting and well analysed.   

We woul like to thank Reviewer 2 for her/his valuable comment. We have revised the entire text to ensure more clarity.

Reviewer 3 Report

This paper presents a method to detect and classify roads in BIM-GIS using LiDAR data. The paper is interesting since is focused on a novel issue related with the binomial Roads-BIM. However, authors should consider the following aspects before the paper can be published. 

1) In section 2, authors should consider other relevant and novel works related with automatic road extraction methods such as: "An automated approach to vertical road characterisation using mobile LiDAR systems: Longitudinal profiles and cross-sections", "Automatic inventory of road cross‐sections from mobile laser scanning system", "Semiautomatic extraction of road horizontal alignment from a mobile LiDAR system" and "Road safety evaluation through automatic extraction of road horizontal alignments from Mobile LiDAR System and inductive reasoning based on a decision tree". Please, include these references as part of the overview presented in section 2.

2) In terms of point cloud processing, the authors outlined a method to classify the road / vegetation / buildings, and then calculate the center of the road, using one of the ISPRS benchmarks. It would be convenient to highlight the main contribution of this processing/classification step. 

3) Regarding the BIM-GIS approach, the use of Autodesk InfraWorks is interesting since provides a powerful approach to infrastructures. However, for getting the final results authors include a lot of manual error correction work which does not give information regarding the time required.

4) The final contribution is focused on the integration of the GIS vector layers into BIM-oriented software. Although, there is no doubt that it is interesting, it does not go into existing models or try to adapt to standards such the current IFCAlignment. Authors should reference these aspects, at least, in future perspectives.

Author Response

We would like to thank Reviewer 3 for her/his valuable comments and suggestions. We took all of them into consideration in the revised version of the manuscript. Such corrections have improved the quality of the paper. We report in this document the original comments and our answers.

This paper presents a method to detect and classify roads in BIM-GIS using LiDAR data. The paper is interesting since is focused on a novel issue related with the binomial Roads-BIM. However, authors should consider the following aspects before the paper can be published. 

1) In section 2, authors should consider other relevant and novel works related with automatic road extraction methods such as: "An automated approach to vertical road characterisation using mobile LiDAR systems: Longitudinal profiles and cross-sections", "Automatic inventory of road cross‐sections from mobile laser scanning system", "Semiautomatic extraction of road horizontal alignment from a mobile LiDAR system" and "Road safety evaluation through automatic extraction of road horizontal alignments from Mobile LiDAR System and inductive reasoning based on a decision tree". Please, include these references as part of the overview presented in section 2.

We agree with the reviewer. We added a short paragraph presenting road extraction works based on car-based LiDAR systems. In the new paragraph we cited previous works suggested by the reviewer and other relevant novel works.

2) In terms of point cloud processing, the authors outlined a method to classify the road / vegetation / buildings, and then calculate the center of the road, using one of the ISPRS benchmarks. It would be convenient to highlight the main contribution of this processing/classification step. 

We agree with the reviewer. Compared to previous research on road centreline extraction we believe that main contributions of our work can be summarized as:

  • The implementation of a multi-scale approach taking advantages of point cloud information at local and global scale allowing the identification of roads (both main roads and dirty roads) characterized by significantly different widths; 
  • The development of a voting scheme allowing to identify and discriminate between roads and other surfaces (parking lots, squares, etc.) characterized by similar local features (local planarity, LiDAR intensity, etc.) in the point cloud; and      
  • The implementation of a regularization procedure allowing for the generation of a road network presenting consistent topological relationship extending to centreline extraction the procedure based on cell complex presented in (Previtali, M., Díaz-Vilariño, L., Scaioni, M. Indoor building reconstruction from occluded point clouds using graph-cut and ray-tracing. Applied Sciences, 8(9), 1529, 2018).

In addition, the paper presents the development of a new procedure for building footprint regularization. 

We added this discussion in Section 4 “Generation of road GIS layers from LiDAR data”

3) Regarding the BIM-GIS approach, the use of Autodesk InfraWorks is interesting since it provides a powerful approach to infrastructures. However, for getting the final results authors include a lot of manual error correction work which does not give information regarding the time required.

We agree with the reviewer. The description provided in the previous version was not sufficiently clear, we could say that defining manual corrections like “simple” is rather superficial. We completely revise this part of the text. We added a better explanation to consider the problem of manual correction as an effective challenge, instead of saying that they can be easily corrected with manual measurements. We believe automation in error detection has to be considered as an effective challenge for future research. We mentioned the solution available in BIM software for buildings, such as interference detection. A similar approach could be extended in the case of infrastructure elements. Then, we tried to clarify that the problem is related to two aspects: (1) identification of an error and (2) correction. Step 1 is completely manual, at the moment we just perform a visual inspection, and we clarified that this is a significant limitation for “real” projects in large areas. Step 2 can be partially automated using the parametric capability of BIM-GIS elements. We think that the description in the revised text is clearer and more concise now.

4) The final contribution is focused on the integration of the GIS vector layers into BIM-oriented software. Although, there is no doubt that it is interesting, it does not go into existing models or try to adapt to standards such the current IFCAlignment. Authors should reference these aspects, at least, in future perspectives.

Yes, this is a fundamental point. We discussed the need for integration with the current standard in the conclusions. We agree with the reviewer that it is better to describe this point and present it as a possible future development. 

Round 2

Reviewer 1 Report

The reviewed version of the manuscript has improved considerably the quality of the work, now the actual limitations of the process and the contributions/extent of the work are clearly specified, which answers to the main issues found in the first version. Minor issues were also corrected and taken care of. 

I do not have further comments on this manuscript.